# Spin–orbit torque switching in a T-type magnetic configuration with current orthogonal to easy axes

W.J. Kong[1], C.H. Wan[1], X. Wang[1], B.S. Tao[1], L. Huang[1], C. Fang[1], C.Y. Guo[1], Y. Guang [1], M. Irfan [1] & X.F. Han[1,2,3]

Different symmetry breaking ways determine various magnetization switching modes driven by spin–orbit torques (SOT). For instance, an applied or effective field parallel to applied current is indispensable to switch magnetization with perpendicular anisotropy by SOT. Besides of this mode, here we experimentally demonstrate a distinct field-free switching mode in a T-type magnetic system with structure of MgO/CoFeB/Ta/CoFeB/MgO where a perpendicular layer with tilted easy axis was coupled to an in-plane layer with a uniaxial easy axis. Current was applied orthogonal to both easy axes and thus also normal to an in-plane effective field experienced by the perpendicular layer. Dynamic calculation shows perpendicular layer could be switched at the same time as the in-plane layer is switched. These field-free switching modes realized in the same T-type magnetic system might expedite the birth of multi-state spin memories or spin logic devices which could be operated by all electric manners.

[1] Beijing National Laboratory for Condensed Matter Physics, Institute of Physics, Chinese Academy of Sciences, Beijing 100190, China. [2] Center of Materials Science and Optoelectronics Engineering, University of Chinese Academy of Sciences, Beijing 100049, China. [3] Songshan Lake Materials Laboratory, Dongguan, Guangdong 523808, China. Correspondence and requests for materials should be addressed to X.F.H. (email: xfhan@iphy.ac.cn)

Spin–orbit torque (SOT)-induced magnetization switching in ferromagnet/heavy metal bilayers can be realized in three modes, Type-x, Type-y, and Type-z, categorized by the relative orientation between polarization of spin current ($\sigma$) and easy axis (EA) of ferromagnet[1]. Here, $\sigma$ is generated by spin–orbit coupling phenomena such as spin Hall Effect[2–6] and/or Rashba–Edelstein effect[7,8]. Supposing current $I_x$ is applied, $\sigma$ polarizes to the $y$-axis when vertically flowing from heavy metal into ferromagnet. In Type-y mode, polarization of $\sigma$ is parallel to EA. In this case spin dynamics is similar to the one induced by classic spin-transfer torques[9–13]. In Type-z mode, $\sigma$ polarizes perpendicular to EA. Then an applied or effective field ($B_{eff}$) along the $x$-axis is indispensable to break structural inversion symmetry and realize deterministic switching. Wedge structures[14], exchange bias[15–18], or interlayer coupling[19,20] have been adopted to introduce $B_{eff}$ and achieve field-free switching of perpendicular magnets.

Switching behaviors of interlayer coupling structures with so-called T-type magnetic configuration are especially interesting. In these structures there exist two magnetic layers, one with perpendicular anisotropy (PMA) and the other with in-plane anisotropy (IMA). SOT-switching chirality (clockwise or counterclockwise in magnetization vs. current curves) of the PMA layer can be controlled by orientation of the IMA layer[19–21]. In this case, current has to be applied parallelly with the EA of the IMA layer and orthogonally to the EA of the PMA layer. Thus the switching still belongs to Type-z mode which is the only reported SOT-manner to switch perpendicular films by now. Here, we will demonstrate an entirely different mode, referred to as Type-T mode, to switch a T-type structure of CoFeB/Ta/CoFeB with proper tilting EA of the PMA layer, in which current is applied normal to both EAs of PMA and IMA layers. As a result, both CoFeB layers can be switched simultaneously by the current and furthermore the switching chirality of the PMA layer is immune to external field. This distinct manner may greatly expand functions and controllability of a SOT device with the T-type magnetic structure.

## Results

**Symmetric analysis and macrospin simulation.** Figure 1a–c shows the measurement setups of Type-y, Type-z, and Type-T modes, respectively. For the mode of Type-T, to be specific, spins of PMA and IMA layers in a T-type structure are configured as shown by Region I of Fig. 1d and $I_x$ is applied along the +x- axis and orthogonal to both EAs. Bearing in mind that magnetization ($M$) is a pseudovector[14,22], then one can easily obtain the other two setups in Region II and III by xoz and yoz mirror symmetrical operations, respectively.

Comparison between Region I and III indicates possibility of switching both IMA and PMA layers by reversing $I_x$. Before this switching becomes achievable, the xoz symmetry has to be broken because Region I and II indicate opposite perpendicular magnetization components ($M_z$) of the PMA layer under the same $I_x$, as indicated by red arrows, thus preventing its deterministic switching[14,22]. This xoz mirror symmetry can be broken by EA tilting of the PMA layer toward EA of the IMA layer (red solid and dashed lines), because this symmetry leads to opposite EA titling of the PMA layer while a real sample only support one tilting direction if the sample is dominated by uniaxial anisotropy (Fig. 1d). Therefore, symmetry breaking law suggests field-free switching takes place only between Region I and III in T-type structure with tilting EA of PMA layer.

We then simulated spin dynamics of the structure in Type-T mode using a macrospin model, taking both PMA and IMA, antiferromagnetic coupling of the two layers and the EA tilting

into account (see Supplementary Note 3 and Supplementary Figs. 7–14). Figure 1e, f show indeed the PMA ($M_1$) and IMA ($M_2$) layers can be simultaneously switched at the same critical current. Tilting direction controls switching chirality of perpendicular components ($M_{1z}$) of $M_1$ vs. $\sigma$ loops. Furthermore, simulation results suggest the EA tilting and interlayer coupling are both indispensable in this switching mode. Spin configurations before and after the switching are shown as insets.

We have also simulated Type-z switching mode with current applied along the EA of the IMA layer (see Supplementary Note 2 and Supplementary Figs. 2–5). Corresponding results (Fig. 1g, h) show typical features of Type-z mode. Switching chirality is determined by orientation of the IMA layer. During switching process, the PMA layer reverses its perpendicular component while the IMA layer retains its in-plane component. This scenario is remarkably distinguished from Type-T mode where both ferromagnetic layers switch simultaneously. Interestingly, $M_{1z}$ vs. $\sigma$ curve shows noticeable offset in Type-z mode, which is caused by the EA tilting of the PMA layer and can be used as a marker to evidence the EA tilting (see Supplementary Fig. 6).

**Device preparation and Type-z switching mode driven by SOT.** Figure 2a shows $M$–$H$ hysteresis loops in the different axes of the film with the structure of Ta(5)/MgO(2)/$Co_{20}Fe_{60}B_{20}$(1.0)/Ta (1.3)/$Co_{20}Fe_{60}B_{20}$(1.4)/MgO(2)/Ta(3) (number in nanometers). Hysteresis was observed in the y- and z- directions while only smooth saturation process was observed in the x- direction, which indicated one CoFeB layer had PMA and the other CoFeB layer had IMA with an in-plane EA along the $y$-direction. The two CoFeB layers were deposited following different growth sequences and had different thicknesses and thus probably exhibited different magnetic anisotropies[23]. We supposed that the bottom and top CoFeB layers had PMA and IMA, respectively. The PMA could also be inferred from sizable anomalous Hall resistance at zero field. Ta, as a spacer between the CoFeB layers, has been reported to mediate strong antiferromagnetic interlayer coupling below 2.0 nm[24], indicating the coupling in our case was dominated by exchange effect instead of dipolar interaction. Here, a similar interlayer coupling (IC) was also evidenced by the following SOT-switching data. Afterward, the raw film was patterned into ellipses with size of 10 μm × 30 μm. The long axis in the $y$-direction was parallel to the EA of the in-plane layer. Four terminals were connected to the ellipses to form Hall bars as shown in Fig. 2b.

SOT-switching in Type-z mode was first checked. Current and EA of the IMA layer were both along the $y$-axis and a field $H_y$ in the same direction was also applied. Hall resistance $R_{xy}$ was used to characterize $M_z$ of the sample. Figure 2c shows typical switching loops ($R_{xy}$ vs. $I$ curves) for a perpendicular film, clockwise and counterclockwise for negative and positive fields, respectively. Increase in $H_y$ reduced critical switching current and squeezed the SOT-switching loops as reported before[3,25]. More importantly, reducing higher field of ±300 Oe than in-plane coercivity to zero, then measuring switching performance as $H_y$ = 0, we found field-free switching could also be realized. Furthermore, switching chirality depended on history of reducing the field, counterclockwise (clockwise) as $H_y$ was reduced from +300 Oe (−300 Oe) to 0 (Fig. 2d). This phenomenon strongly evidenced IC between the perpendicular and the in-plane layers. Without the IC, the perpendicular film would not have been switchable only by SOT and switching chirality in field-free condition would not have depended on the state of the in-plane layer. We have estimated resistivity of Ta about 219 μΩcm supposing current density uniformly distributed in the Ta layers

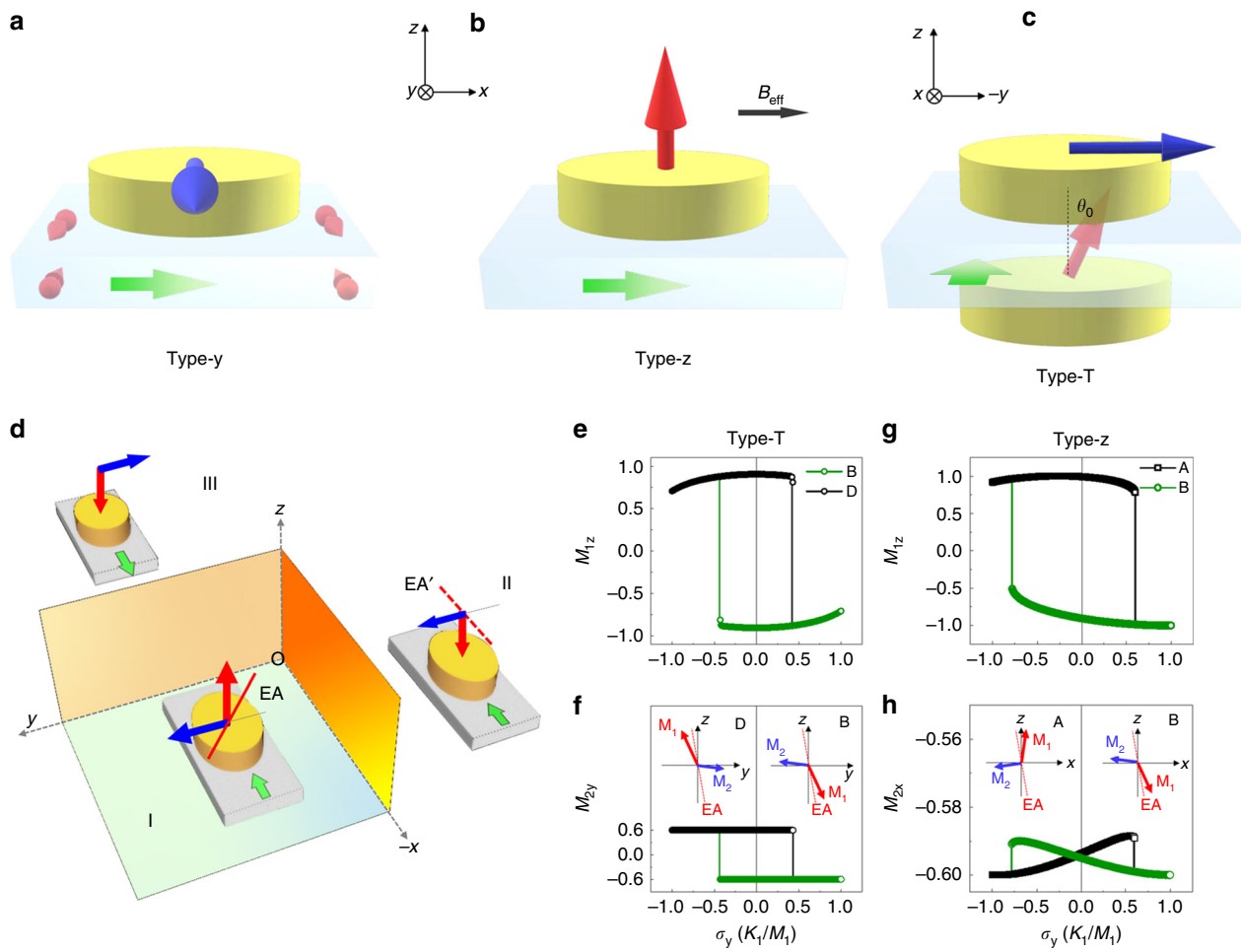

**Fig. 1** Different SOT-switching modes. **a–c** Type-y, Type-z, and Type-T mode. Green, red, and blue arrows denote current, magnetization of PMA, and IMA layers, respectively. **d** Symmetry breaking in Type-T mode. A tilting EA of PMA indicated by red solid or dashed line is necessary. **e**, **f** Simulated $\sigma$ dependence of $M_{1z}$ and $M_{2y}$ in Type-T mode in which current is vertical to EA of the IMA layer. **g**, **h** Simulated $\sigma$ dependence of $M_{1z}$ and $M_{2x}$ in Type-z mode in which current is parallel to EA of the IMA layer. Spin configurations before and after switching are shown as insets

and two CoFeB layers. We have also calibrated SOT efficiency in the other Ta(5)/MgO(2)/CoFeB(1.2)/Ta(1.3)/CoFeB(1.2)/MgO (2)/Ta(3) stack with double perpendicular layers by harmonic measurement[25,26]. Obtained damping-like and field-like torque efficiencies were about 0.60 nT per A cm$^{-2}$ and 0.74 nT per A cm$^{-2}$, respectively. These values were comparable with the previous ones[1,2,27], indicating Ta in this system could also function as an efficient SOT supplier. Though IC strength was still not strong enough to assist full SOT-switching and worthy of optimizing in the future, field-free SOT switching realized in the Ta/CoFeB/MgO structure promised its compatibility with mainstream technology of perpendicular magnetic random access memory (p-MRAM) in which the structure has already emerged as a basic building block.

More importantly, a sizable offset in $R_{xy}$ vs. $I_y$ occurred (Fig. 2c, d), which strongly evidenced the EA tilting of the PMA layer as shown in Fig. 1g, h. In contrast, if there were no tilting, Type-z switching loops should have been symmetric about the axis of $I_y = 0$ as reported in the single PMA layer[3,25] or in similar T-type structures[20,21]. By comparing degree of offset in experimental and theoretical results, we estimated that the EA of the PMA layer tilted by 10° toward y- axis (see Supplementary Fig. 6). This EA tilting was likely to be induced by high-temperature annealing with an in-plane field of 0.7 T. This special T-type structure with

tilting EA of PMA layer endowed us with chance of realizing the proposed Type-T mode for SOT switching.

**Type-T switching mode of SOT.** Then as shown in Fig. 3a we patterned a similar type of Hall bars but with the long axis of the ellipse vertical to the EA of the in-plane layer. Though shape anisotropy would change the magnitude of total anisotropic energy, it is noteworthy that the EA of the in-plane layer still remained along the y- axis, which had been inferred from anisotropic magnetoresistance measurement. Figure 3b shows a typical $H_z$ dependence of $R_{xy}$. Full switching of the perpendicular layer resulted in $\Delta R_{xy}$ of 50 mΩ. Afterward, we measured switching curves with both current and field along the x- axis while EA of the IMA layer was along the y- axis. The most striking behavior was nearly 100% switching degree realized in the field-free condition. Furthermore, $H_x$ of 300 Oe with opposite signs could not change switching chirality in this Type-T mode any more (Fig. 3c). Even as higher fields of above ±1 kOe were applied, the switching chirality was still not changed. However, the field parallel to current changed the switching chirality in the Type-z mode (Fig. 2d). It was a remarkable difference between the two modes (see Supplementary Table 1). The insensitivity of Type-T mode to $H_x$ was

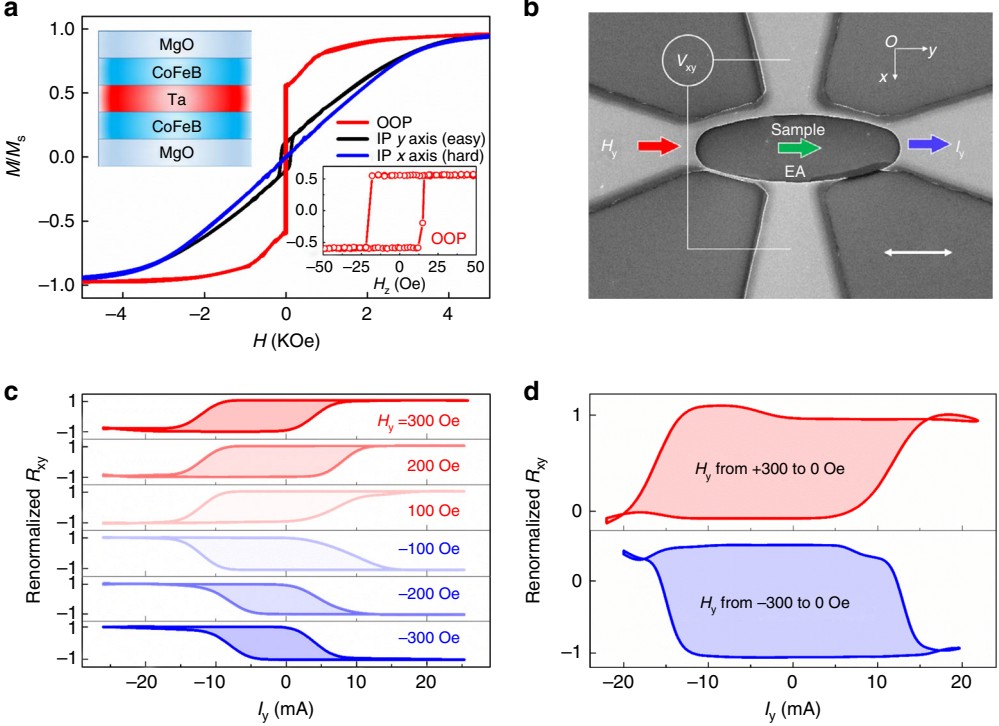

**Fig. 2** Field-free switching in Type-z mode. **a** M–H hysteresis of the raw film. **b** Structure of a patterned film. Scale bar: 10 μm. **c** Field assistance magnetization switching driven by current ($I_y$). **d** Field-free switching of the Hall bar. Renormalized $R_{xy} \equiv (R_{xy} - R_O)/\Delta R_{xy}^{max}$. $R_{xy} = V_x/I_y$, $R_O$ was the median of the $R_{xy}$ vs. $H_z$ curve and $\Delta R_{xy}^{max} = |R_{xy}(H_z = 0) - R_O|$

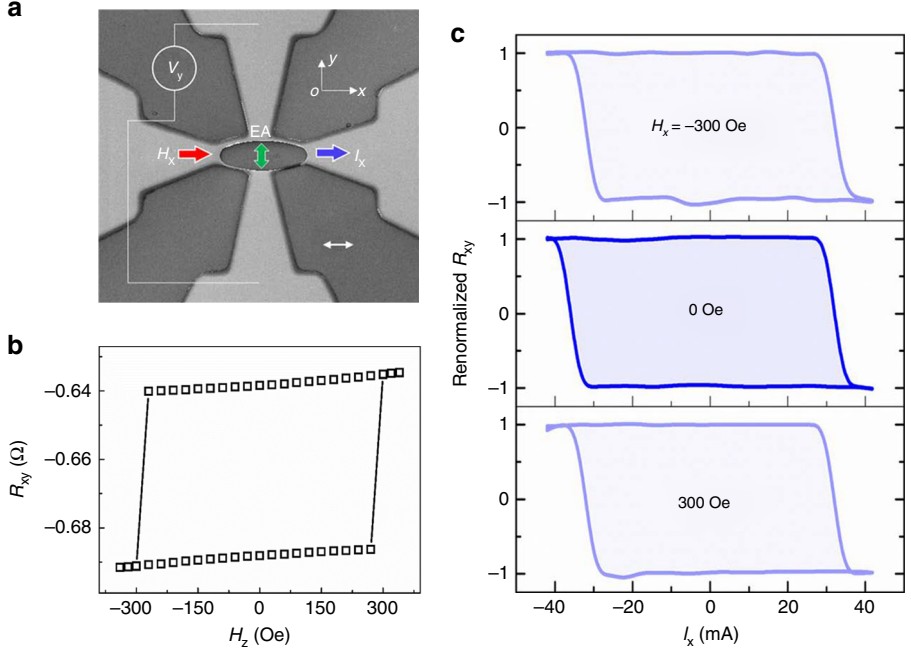

**Fig. 3** Characteristics of Type-T mode. **a** Scanning electron microscope image of a Hall bar device. Scale bar: 10 μm. **b** $H_z$ dependence of $R_{xy}$ of the device. **c** Current dependence of renormalized $R_{xy} = (R_{xy} + 0.663\,\Omega)/0.025\,\Omega$ under different field $H_x$ which is parallel to current

reproduced in the simulation (see Supplementary Note 4 and Supplementary Fig. 13). We have also used opposite current scanning methods to measure Fig. 3c. Results were the same for different scanning orders (see Supplementary Note 1 and Supplementary Fig. 1). Circular devices were also tried. They showed similar switching behaviors.

Simulated switching curves in Fig. 1e, f show $M_{1z}$ and $M_{2y}$ switches simultaneously at the same critical torque. Spin dynamic analysis on Type-T mode (see Supplementary Fig. 12) further shows that $M_{1z}$ is switched by the joint action of antiferromagnetic coupling with IMA layer and SOT as $M_{2y}$ is switched by SOT. Since $H_y$ can offset $M_{2y}$ switching by giving IMA layer an

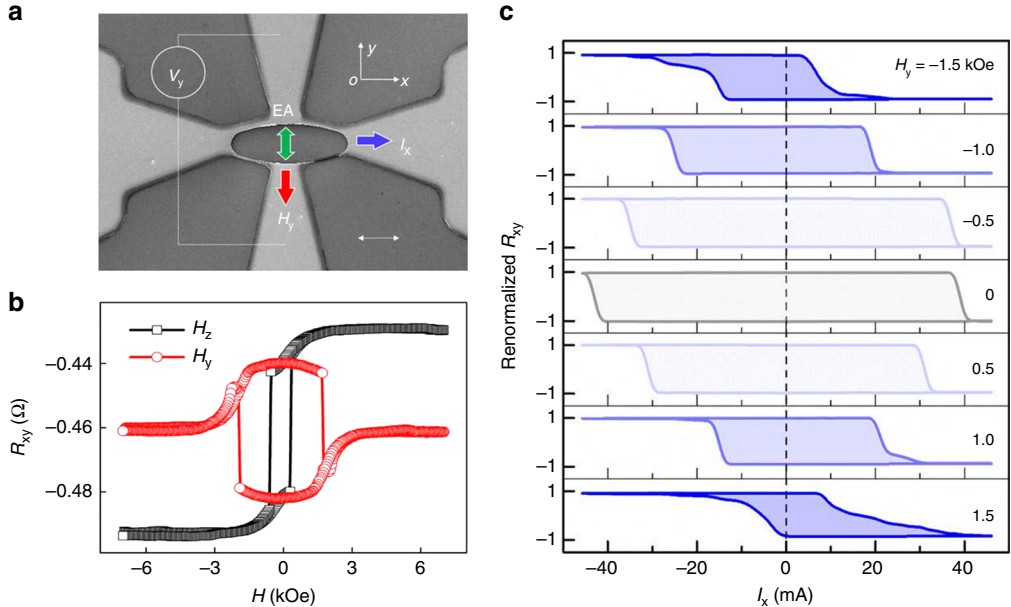

**Fig. 4** Switching characteristics as applying a bias field along the EA. **a** Scanning electron microscopy image of the measurement setup. Scale bar: 10 μm. **b** $R_{xy}$ as functions of $H_z$ and $H_y$. **c** $I_x$ dependence of renormalized $R_{xy} = (R_{xy} + 0.461\,\Omega)/0.020\,\Omega$ under different $H_y$

additional unidirectional anisotropy, it is nature to infer that $H_y$ can also offset $M_{1z}$ switching in this Type-T mode.

As shown in Fig. 4a, we fabricated another ellipse device. Its long axis was normal to the EA, current was in the x- direction and field was applied in the y- axis and parallel to the EA. Before SOT-switching measurement, sample was tilted by balancing $R_{xy}$ (6 kOe) and $R_{xy}$ (−6 kOe) to ensure the field parallel to the y axis and there was little $H_z$ component. A clear transition in $R_{xy}$ vs. $H_y$ curves (Fig. 4b) was observed after $H_y$ had been calibrated in-plane, which also indicated a tilting EA of the PMA layer. Full switching of the perpendicular layer resulted in $\Delta R_{xy}$ of about 40 mΩ.

As shown in Fig. 4c, $H_y$ with different polarities and magnitudes could not change the switching chirality (clockwise here). However, $H_y$ with different polarities indeed led to opposite offsets of the switching loops as expected. If there was no IC between the two layers, $I_x$ would not have switched the perpendicular layer even though $H_y$ was applied, which also evidenced the existence of IC. Another function of $H_y$ was to tilt the perpendicular layer away from the z-axis, which could lower energy barrier for the switching of this PMA layer and reduce its critical switching current as occurred in Type-z mode[3,25]. Offset of $M_{1z}$ vs. σ loops by $H_y$ were reproduced (see Supplementary Fig. 14). $R_{xy}$-$I$ curve still shows switching behaviors at very high $H_y$ of ±1.5 kOe, which can suppress the switching of the IMA layer. The switching in this field region cannot be of Type-T mode. Noticeably, different from those sharp switching at relative low field, the switching at high $H_y$ is very gentle, possibly due to continual nucleation as reported by Miron et al.[28].

## Discussion

Realization of two different switching modes in the same MgO-based MRAM-compatible structure could bring about some application merits. First, Type-T and Type-z modes have different field dependence. The former is not sensitive to external field while the latter is, indicating Type-T mode might be more suitable to construct stable SOT-MRAM devices with strong robustness against field turbulence. Second, the same structure

has two different operational methods, which provides chances to design more functional devices. Third, as our proposal regarding Type-T switching shown, the IMA and PMA layers could be coherently controlled by the same critical current while only the PMA layer could be switched in Type-z mode, which made the structure possible to construct multistate memories.

In this study, we have applied a macrospin model to interpret Type-T switching characteristics. Actually, another model based on SOT-driven domain wall motion was also proposed before[29]. The model requires two conditions: (1) Dzyaloshinskii-Moriya interaction induced Néel-type domain wall and (2) effective perpendicular field experienced by the domain wall $\mathbf{H}_{eff}$ (proportional to $\mathbf{m}_{DW} \times \mathbf{s}$) as shown in Fig. 1 of ref. [29]. $\mathbf{m}_{DW}$ is the center moment of a domain wall and $\mathbf{s}$ is absorbed spin current by the domain wall. In Type-z mode, field parallel with $I$ aligns $\mathbf{m}$ normal to $\mathbf{s}$. $\mathbf{H}_{eff}$ can be thus along the z-axis. However, exchange coupling field experienced by the perpendicular layer is normal to $I$ in Type-T mode, which inclines to align the $\mathbf{m}_{DW}$ parallel with $\mathbf{s}$. $\mathbf{H}_{eff}$ would become 0. The domain wall motion model fails to account for the behavior here. Nevertheless SOT can still contribute to nucleation in incoherent switching process[30]. The difference between this multidomain switching process and macrospin model is the anisotropic energy, which should be replaced by an effective anisotropic energy of some local nucleation sites instead of the average one of the film. The macrospin model can therefore qualitatively help one to understand the switching behaviors in Type-T mode.

In summary, besides of Type-z mode, a different type of field-free switching mode (Type-T mode) was discovered in the MgO/CoFeB/Ta/CoFeB/MgO system with T-type magnetic configuration. This mode could enable a current applied along the in-plane hard axis of the in-plane layer to switch both perpendicular and in-plane layers at the same critical current. And the switching chirality of the perpendicular layer in this mode was relatively inert to external fields, compared with Type-z mode. Besides, the core structure MgO/CoFeB/Ta/CoFeB/MgO was compatible with current mainstream technology of p-MRAM. Realization of rich

SOT-switching modes, both Type-T and Type-z modes, could advance the development of field-free SOT-MRAM and even endow community probabilities to construct versatile spin logic or multistate spin memory devices that are fully operational via ideal electrical manners.

## Methods

**Materials and device fabrication**. The film was deposited on 4-inch Si/SiO$_2$ (500 nm) wafers. The stack with T-type magnetic configurations was substrate//Ta(5)/ MgO(2)/Co$_{20}$Fe$_{60}$B$_{20}$(1.0)/Ta(1.3)/Co$_{20}$Fe$_{60}$B$_{20}$(1.4)/MgO(2)/Ta(3 nm) deposited by Semiconductor R&D Sputtering system-ROTARIS (Singulus Technologies AG). Then a two-step annealing process was implemented inside a high vacuum chamber of $4.1 \times 10^{-4}$ Pa. Firstly, the stacks were annealed at 275 °C for 1 hour with a perpendicular magnetic field of 0.7 T and then, the same stacks were annealed at 310 °C for 20 minutes with the field along the film plane to enhance in-plane anisotropy and tilt the PMA layer. Another stack substrate//Ta(5)/MgO (2)/Co$_{20}$Fe$_{60}$B$_{20}$(1.2)/Ta(1.3)/Co$_{20}$Fe$_{60}$B$_{20}$(1.2)/MgO(2)/Ta(3 nm) with two perpendicular CoFeB layers was used to characterize SOT-efficiencies. The magnetic properties were obtained via vibrating sample magnetometer (micro sense EZ-9). Then the stacks with T-type magnetic configuration was patterned into Hall bar devices in the following three steps: First, no useful area was etched out to SiO$_2$ substrate with the working ellipse area and the electrode areas retained. Second, the ellipse working area was covered by resist and the electrode areas were etched to the bottom MgO layer. In this case the IMA layer in the electrode areas were also all etched out. Third, Au electrodes were deposited to cover the etched electrode areas by sputtering and lift-off technology. Thus current can reach the CoFeB/Ta/CoFeB sandwich through side walls of a Hall device.

**Device measurement**. The transport properties were measured in probe station where magnetic field was generated via Helmholtz coils free of ferromagnetic cores. Thus the zero field condition (<0.5 Oe) could be firmly realized. A larger field than 300 Oe was provided by a standard Physical Property Measurement System (Quantum Design). SOT-switching performance was obtained by measuring current dependence of Hall resistance in a nonvolatile method[25]. Current was sourced by a Keithley 2400 and Hall voltage was picked up by a Keithley 2182 nanovoltmeter. It is worth noting only 180° switching of the perpendicular layer could be directly detected by anomalous Hall effect while 180° switching of the in-plane layer could only be inferred from the switching of the perpendicular layer in Type-T mode due to incompetence of planar Hall effect to monitor 180° switching of the in-plane layer in our Hall bar devices. All the measurement were conducted at 300 K.

## Data availability

The authors declare that the data supporting the findings of this study are available from the corresponding author upon reasonable request.

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

## Acknowledgments

This work was supported by the National Key Research and Development Program of China (MOST) (Grant nos. 2017YFA0206200 and 2018YFB0407600), the National Key Scientific Instrument and Equipment Development Projects (MOST) [Grant no. 2011YQ120053], the National Natural Science Foundation of China (NSFC) [Grant nos. 11404382, 11434014, and 51620105004], the Strategic Priority Research Program (B), the Key Research Program of Frontier Sciences and the International Partnership Program of Chinese Academy of Sciences (CAS) [Grant nos. XDB07030200, QYZDJ-SSW-SLH016, and 112111KYSB20170090].

## Author contributions

X.F.H. led and was involved in all aspects of the project. C.H.W. and W.J.K. were the first-authors contributed equally to this work. X.F.H and Singulus Technologies AG fabricated stacks using Semiconductor R&D Sputtering—ROTARIS. W.J.K, C.F., and L. H. annealed stacks. W.J.K, C.Y.G., Y.G., and X.W. fabricated microsized Hall devices. B. S.T., W.J.K., and M.I. conducted magnetic property measurement. W.J.K. and C.H.W. conducted switching and transport measurements. C.H.W., W.J.K., and X.F.H. contributed to macrospin modeling and wrote the paper. X.F.H. and C.H.W. supervised and designed the experiments. All the authors contributed to data mining and analysis.

## Additional information

**Competing interests:** The authors declare no competing interests.

