## [Peer Review File · Nature Communications]

Reviewers' comments:

Reviewer #1 (Remarks to the Author):

The authors reported an observation of the magnetization switching by using SOT effect through a SOT channel, based on a stack MgO/CoFeB/Ta/CoFeB/MgO with one magnetic layer perpendicular and one magnetic layer in-plane. The topic is timely and interesting. There are several concerns related with the results presented in the manuscript.

1. Although the experimental results agree well with the simulation from macro spin model (in the Supplemental), I still don't quite understand the switching mechanism of the PMA layer. For a single layer PMA on a spin Hall channel, it wouldn't be switched under a current I_x and field H_y (field normal to current). Then why in this paper, the PMA layer can be switched under a current I_x and exchange coupling field along y (exchange field normal to the current)?
2. In this paper, only exchange coupling between the two layers was considered. It is very likely that dipole coupling also exists in this structure. The in-plane component of the dipole field from the IMA layer to PMA layer would behave similarly with the anti-ferromagnetic exchange coupling field. Can the authors characterize the strengths of both exchange coupling and dipole coupling in their devices?
3. Did the authors study devices with different Ta thicknesses? Maybe this could help to distinguish exchange coupling and dipole coupling.
4. I don't see any reason why the authors use ellipse shape. Since the easy axis of the IMA layer doesn't rely on shape, wouldn't a circular pillar be better? With circular devices can the authors get the same results?
5. Since the spin Hall layer (Ta) is in the middle of IMA CoFeB and PMA CoFeB, how the devices were patterned and connected to electrodes? To which layer the channel region was etched? Has the IMA layer in the bottom been etched? Please provide more fabrication details.
6. Putting spin Hall layer in between would also cause huge troubles for applications.
7. In Fig. 2c, all the loops switch between $R_{xy} = -1$ and $R_{xy} = 1$. However, in Fig. 2d, why the red loop only switches between $R_{xy} = 0$ and $R_{xy} = 1$, and the blue loop only switches between $R_{xy} = 0$ and $R_{xy} = -1$ (not full switching)?
8. In Fig. 4c, the offset of the loops at different H_y has been explained as well as validated by simulation. However, the squeezing of the loop at larger H_y hasn't been explained.

Reviewer #2 (Remarks to the Author):

Report on the manuscript "Spin-orbit torque switching in a T-type magnetic configuration with current orthogonal to easy axes" by W.J. Kong et al.

The manuscript deals with experimental measurements of switching property of a T-type magnetic configuration. Those devices are composed by two ferromagnets with different easy axis separated by an heavy metal (HM). The current that flows in the HM gives rise to spin-orbit torque (SOT) that can drive the switching in absence of an external field.

The demonstration of SOT switching is not new, it has been already observed in several experiments. The need of a magnetic field for a SOT switching has been removed by using different strategies such as a proper geometry. The switching mechanism of a T-type geometry

proposed here is interesting as a path to remove the need of the external field while maintaining an elliptical shape for the cross section of the memory cell, but this results is a technical improvement and in my opinion do not represent a significant advance in the field. However, I believe that the main contribution of this work should be oriented to the 4-state memory that can be achieved with this T-type system as claimed by the authors. In other words, the experimental demonstration of 4 states resistance states in those devices will be a significant results that should merit the publication in Nat. Comm.

Main comments

1. Hysteresis loops showing 4 different resistance states should be added in the main text. I suggest to discuss the impact of this result from a technological point of view.
2. The hysteresis loop of the OPP field scan in Fig.2 a is not clearly visible.
3. The resistance of the Ta should be estimated.
4. The spin-Hall angle and the field like torque should be estimated.
5. Which is the current density necessary for the switching at zero bias field? Compare this number with the other switching strategies at zero field.
6. The size of the device is large for application in real magnetic devices, I believe that the switching process is not uniform. However, the macrospin model used for a simplified description of the device properties reproduce qualitatively the experimental behaviour. Add a comment about this aspect.
7. Comment on the contribution to the
8. Add a more detailed description of the contribution to the magneto resistance (MR). The IMA material should give an anisotropic MR while the PMA materials should contribute with the spin Hall resistance.

Technical comment

More details about the spin-Hall geometry should be added, which material and how the current is injected in the stack.

Responses to Reviewer Comments

We first sincerely thank both referees for their insightful and enlightening comments and suggestions that have helped us to improve the clarity and quality of our manuscript. We have already taken all comments into consideration and revised the manuscript accordingly. Below we provide point-to-point response to all comments and questions.

Reviewer #1 (Remarks to the Author):

Remarks: The authors reported an observation of the magnetization switching by using SOT effect through a SOT channel, based on a stack MgO/CoFeB/Ta/CoFeB/MgO with one magnetic layer perpendicular and one magnetic layer in-plane. The topic is timely and interesting. There are several concerns related with the results presented in the manuscript.

Reply: We appreciate the encouraging and insightful comments.

Question 1. Although the experimental results agree well with the simulation from macro spin model (in the Supplemental), I still don't quite understand the switching mechanism of the PMA layer. For a single layer PMA on a spin Hall channel, it wouldn't be switched under a current I_x and field H_y (field normal to current). Then why in this paper, the PMA layer can be switched under a current I_x and exchange coupling field along y (exchange field normal to the current)?

Reply: Indeed, SOT switching is enabled only when inversion symmetry is broken, so that the symmetry-breaking factors, such as external or effective magnetic field in previous papers [Miron et al. Nature 476, 189 (2011); Liu et al. PRL 109, 096602 (2012); Fukami et al. Nat. Mater. 15, 535 (2016); et al.] is indispensable to realize deterministic switching. In these SOT mechanisms, a fixed external or effective magnetic field along the current channel will determine the switching polarity. However, our proposed mechanism does not fall in this realm because the exchange coupling field provided by the in-plane magnetic anisotropic layer (IMAL) is not fixed when considering the fact that the IMAL will also be switchable by applying current normal to the east axis of the IMAL (y_{Easy}).

In detail, we could make a comparison between this SOT mechanism and the previous one. At first, the macrospin model is widely recognized as a tool to qualitatively understand the SOT-dynamics. Utilizing a macrospin model, we could discuss the static and dynamic switching process of the perpendicular magnetized layer (PMAL) in both mechanisms. In the reported Type z mode, when the PMAL reverses from one state to the other, the external/effective field are fixed in the same direction to determine switching polarity. As sketched below in Figure R1 **a**, **b**, the effective field along current channel will remain in the same direction, judging by the result that the moment of the IMAL in its easy axis ($m_{y_{\text{Easy}}}$) maintains the same sign as the moment of the PMAL (m_z) reverses. So this mechanism relies on the fixed field to break inversion symmetry.

However, when the effective field induced by IMAL is transverse to current channel, as we proposed here, the m_{y_Easy} will switch to another direction and change its sign in response to I when m_z of PMAL switches (Figure R1 **c**, **d**). This switchable layer becomes the symmetry-breaking key which leads to deterministic switching of the PMAL. This phenomenon could be understood in static analysis. As we discussed in the revised manuscript, by increasing I , the Type-y mode ensures the m_{y_Easy} to switch, and then m_z of PMAL reverses to satisfy symmetry-breaking law. Thus switching of both IMAL and PMAL leads to monostable state with increasing current. Dynamic simulation in Figure R1 **c**, **d** have shown the feature that the m_z of PMAL can be switched and also m_{y_Easy} switches.

In conclude, this Type-T mode utilize a switchable IMAL to induce a switchable in-plane field (normal to current) while the previous SOT mechanism relies on a fixed effective/external field (parallel to current) to break symmetry, which is the core concept of Type-T mode. To make it clear, we have added the discussion regarding the dynamic analysis into the supplementary information.

Figure R1 **a** and **b**, switch dynamics of the reported Type z mode with a fixed effective field along the current channel. **c** and **d**, switch dynamics of the Type T mode here with a switchable effective field normal to the current channel. **a** and **c** show the time-evolution of the in-plane magnetic anisotropic layer (IMAL) while **b** and **d** show that of the perpendicular magnetized layer (PMAL).

Question 2. In this paper, only exchange coupling between the two layers was considered. It is very likely that dipole coupling also exists in this structure. The in-plane component of the dipole field from the IMA layer to PMA layer would behave similarly with the anti-ferromagnetic exchange coupling field. Can the authors characterize the strengths of both exchange coupling

and dipole coupling in their devices?

Reply: Before we designed and ordered the $\text{Co}_{20}\text{Fe}_{60}\text{B}_{20}/\text{Ta}/\text{Co}_{20}\text{Fe}_{60}\text{B}_{20}$ stacks from Singulus Technologies AG, we have referenced a paper from Cheng et al. [J. Appl. Phys. 112, 033917 (2012)] which had systematically studied the coupling effect in the $\text{Co}_{20}\text{Fe}_{60}\text{B}_{20}/\text{Ta}(t)/\text{Co}_{20}\text{Fe}_{60}\text{B}_{20}$ and $\text{CoFeB}/\text{Ta}/\text{Ru}/\text{Ta}/\text{CoFeB}$ systems. As shown in Figure R2 from the reference, clear coupling field oscillation as function of spacer thickness had been observed with period of 1.3 nm and the first peak for AFM coupling occurred at 1.2~1.3 nm. It evidenced that exchange coupling effect would be dominant over dipolar interaction in this system. Besides, we have also tried $\text{CoFeB}/\text{Ta}(1.2\text{ nm})/\text{CoFeB}$ stacks with double perpendicular layers in which AFM is also clearly observed [Shi, Phys. Rev. B, 95, 104435 (2017)]. In current stage, we could not quantitatively quantify the contributions from exchange coupling and dipolar interaction. But according to the literature we think exchange coupling field would probably be also dominant though the contribution from dipole interaction could not be fully ruled out. We have also tried another Ta thickness of 1.0 nm. However, it is pity that the ideal T-type magnetic structure was not realized in that system (not shown in the manuscript). We have revised manuscript accordingly regarding the analysis on the nature of interlayer coupling. Worth pointing, this revision will not influence our analysis on the switching modes.

Figure R2 Ta thickness leads to oscillatory interlayer exchange coupling interaction. Figure from Ref. [Cheng, et al. J. Appl. Phys. 112, 033917 (2012)].

Question 3. Did the authors study devices with different Ta thicknesses? Maybe this could help to distinguish exchange coupling and dipole coupling.

Reply: We agree that Ta play an important role in inducing strong interlayer exchange coupling and sourcing SOT. Plenty of research efforts has been donated to reveal the influence of Ta spacer, especially in structure of $\text{CoFeB}/\text{Ta}/\text{CoFeB}$. By tune the thickness of spacer layer, the exchange coupling could oscillate in ferromagnetic, zero and antiferromagnetic [Parkin et al. Phys. Rev. Lett. 64, 2304 (1990); Parkin et al. Phys. Rev. Lett. 67, 3598 (1991); Cheng et al. J. Appl. Phys. 112, 033917 (2012)]. As for Ta spacer layer, Cheng et al [J. Appl. Phys. 112, 033917 (2012)] reported that $\text{CoFeB}/\text{Ta}/\text{CoFeB}$ structures have shown a parallel-antiparallel oscillatory behavior. The oscillatory behavior results from the interlayer exchange interaction and varies as a function of the Ta spacer thickness with a period of ~1.3 nm, as shown in the above Figure R2 from the reference paper, confirming that interlayer exchange coupling dominate coupling behavior, other than dipolar

interaction. Moreover, we intentionally chose the Ta thickness to around the first coupling strength peak, at the thickness of ~ 1.3 nm, because strong interlayer coupling will enable the IMAL to induce stronger in-plane effective field and to reduce critical switching current, which is very important to achieve field free magnetization switching. As a SOT source layer, 1.3 nm Ta is also effective to generate sufficient spin current. Actually, we have also tried Ta thickness of 1 nm. However, the T-type magnetic configuration was not realized in the stack. Thus no switching performance was carried out there.

Question 4. I don't see any reason why the authors use ellipse shape. Since the easy axis of the IMA layer doesn't rely on shape, wouldn't a circular pillar be better? With circular devices can the authors get the same results?

Reply: The shape of the device has little influence on anisotropy of the PMAL while affecting magnetic property of the IMAL. As the IMAL directly relates to the switching mechanism, device shape plays some minor role in SOT switching. More specifically, when the long axis of ellipse is collinear with the easy axis of the IMAL, the field-free Type z mode switching is robustly realized but the switching of Type-T mode will be suppressed in a certain degree, because the shape anisotropy increases the barrier against the switching of the IMAL. So we could not achieve Type-T mode in this case. Instead, while the long axis is directed to the hard axis, the shape anisotropy decreases the barrier so that the IMAL would be easier to switch by current. Then Type-T mode is realized. However, in this case, the field-free switching in Type z mode is suppressed to some extent.

We also tried circular devices as well. The film was patterned into circular devices with diameter of $5 \mu\text{m}$. When current was applied in the hard axis, field-free magnetization switching in Type T mode is also realized (Figure R3), indicating the shape indeed does not play a critical role in this Type T mode as pointed out by the referee. We have revised the manuscript accordingly by adding a comment "We have also tried circular devices which showed similar switching behaviors in Type T mode" after description of Figure 3.

Figure R3 **a**, SEM of $5 \mu\text{m}$ circular bar and **b**, field-free SOT switching in Type T mode for circular Hall device as I is applied along the hard axis.

Question 5. Since the spin Hall layer (Ta) is in the middle of IMA CoFeB and PMA CoFeB, how

the devices were patterned and connected to electrodes? To which layer the channel region was etched? Has the IMA layer in the bottom been etched? Please provide more fabrication details.

Reply: When we pattern the device, three steps are implemented. First, no useful area is etched out to SiO₂ substrate with the working ellipse area and the electrode areas retained. Second, the ellipse working area was covered by resist and the electrode areas were etched to the bottom MgO layer. In this case the IMA layer in the electrode areas were also all etched out. Third, Au electrodes were deposited to cover the etched electrode areas by sputtering and lift-off technology. We have also revised method part accordingly.

Comments 6. Putting spin Hall layer in between would also cause huge troubles for applications.

Reply: We think Ta as a spacer layer sandwiched between two ferromagnetic layers can bring about an important merit as shown below though it sets high requirement in patterning process.

Previous work [Lau et al. Nature Nano. 11, 758 (2016)] use a Ru spacer on one side of a perpendicular magnetized layer and adopt spin Hall Pt channel in the other side of this perpendicular layer. In this structure, Lau showed field-free switching in Type *z* mode, utilizing SOT and exchange coupling. However, this structure is not compatible with sophisticated MTJ technology since the free perpendicular ferromagnetic layer has already been sandwiched by two metal layers so that this free layer fails to directly connect with MgO.

Here we prove that Ta as both spacer layer and spin Hall channel in the CoFeB/Ta/CoFeB system could enable two kinds of field-free switching modes (Type *z* and Type T). Furthermore the structure can be compatible with CoFeB/MgO/CoFeB MTJ technologies, so that realizing field free switching in this structure has strong practical application impact.

Certainly, placing spin Hall layer in-between will set difficulty in etching process. Accurate control on etching thickness will be necessary. Luckily it is feasible for current technology to control the etching thickness in very high accuracy [Henri Jansen et al. J. Micromech. Microeng. 6, 14 (1996)]. Overall, from the application viewpoint, Ta based CoFeB perpendicular structure can still be a promising choice for SOT-MRAM devices.

Question 7. In Fig. 2c, all the loops switch between $R_{xy} = -1$ and $R_{xy} = 1$. However, in Fig. 2d, why the red loop only switches between $R_{xy} = 0$ and $R_{xy} = 1$, and the blue loop only switches between $R_{xy} = 0$ and $R_{xy} = -1$ (not full switching)?

Reply: We appreciate the referee for this insightful comment. Actually, field-free switching in Type *z* mode is not full as pointed by the referee. More strictly speaking, the red loop switches between $R_{xy} = -0.11$ and $R_{xy} = 1$ while the blue loop switches between $R_{xy} = +0.35$ and $R_{xy} = -1$ though field-assistant switching occurs between $R_{xy} = 1$ and $R_{xy} = -1$. Here R_{xy} has been renormalized by the rule described in the manuscript. This partial switching in Type *z* mode is probably because the effective in-plane exchange coupling field is not strong enough to assist full switching. Further enhancement of the interaction, for example, by means of Ta/Ru/Ta [Cheng et al. J. Appl. Phys. 112, 033917 (2012)] or Ta/Ir/Ta spacer, is worthy of trying in the following studies. Though the interaction is still not strong enough to realize full switching in Type *z* mode, it seems strong enough for Type-T switching, which constitutes the main contribution of this work.

As pointed by the referee, to keep description of the experimental data in Figure 2d precise, we have added the following comments into the revised manuscript. “Noticeably, though IC strength was still not strong enough to assist full SOT-switching and worthy of optimizing in the future, field-free SOT switching realized in the Ta/CoFeB/MgO structure promised its compatibility with mainstream technology of perpendicular magnetic random access memory (p-MRAM) in which the structure has already emerged as a basic building block.”

Question 8. In Fig. 4c, the offset of the loops at different H_y has been explained as well as validated by simulation. However, the squeezing of the loop at larger H_y hasn't been explained.

Reply: We appreciate the referee for this insightful comment. The squeezing of the loop results from the increase in H_y . In Figure 4c, the applied H_y along the EA of the IMA layer has two functions. One is to introduce a unidirectional anisotropy to the IMA layer, which leads to offset of the SOT-switching curves in Type T mode as we have simulated by the macrospin model. The other one, we think, is to tilt the perpendicular layer away from the z axis, which can lower energy barrier for the switching of this PMA layer and reduce its critical switching current as occurred in Type-z mode [Lee et al. App. Phys. Lett. 102, 112410 (2013); Zhang, XFH, et al. Phys. Rev. B, 94, 174434 (2016)].

Reviewer #2 (Remarks to the Author):

Report on the manuscript "Spin-orbit torque switching in a T-type magnetic configuration with current orthogonal to easy axes" by W.J. Kong et al.

Remarks: The manuscript deals with experimental measurements of switching property of a T-type magnetic configuration. Those devices are composed by two ferromagnets with different easy axis separated by a heavy metal (HM). The current that flows in the HM gives rise to spin-orbit torque (SOT) that can drive the switching in absence of an external field.

The demonstration of SOT switching is not new, it has been already observed in several experiments. The need of a magnetic field for a SOT switching has been removed by using different strategies such as a proper geometry. The switching mechanism of a T-type geometry proposed here is interesting as a path to remove the need of the external field while maintaining an elliptical shape for the cross section of the memory cell, but this results is a technical improvement and in my opinion do not represent a significant advance in the field. However, I believe that the main contribution of this work should be oriented to the 4-state memory that can be achieved with this T-type system as claimed by the authors. In other words, the experimental demonstration of 4 states resistance states in those devices will be a significant results that should merit the publication in Nat. Comm.

Reply: We appreciate this insightful comment. Realization of 4-states or multistate memory in purely spintronic way is indeed important as pointed out by the referee. Nevertheless, we can only indirectly infer the realization of 4-states and physically cannot directly measure 4-states in the current device as discussed following. Before that, please allow us to address the impact of this study first.

(1) Realization of a new switching mode is scientifically important because it could bring about new spin dynamics and new design of SOT-MRAM elements, we think. Miron [Nature 476, 189 (2011)] and Liu [PRL 109, 096602 (2012)] demonstrated Type-z switching; Liu [Science 336, 555 (2012)] showed Type-y switching; Fukami [Nat. Nano. 11, 621 (2016)] realized Type-x switching. Different switching modes are characterized by different symmetry-breaking manners. Here we used a totally different way, a **switchable** effective field **transverse to current**, to break inversion symmetry and further realize SOT-switching in field-free condition, which has already constituted a new switching mode. To see difference in spin dynamic of Type z and T switching, please refer to our answer to Question 1 of the 1st referee or revised supplementary information.

(2) Meanwhile, we have actually realized two kinds of switching modes, Type z (Figure 2) and Type T (Figure 3) modes, in the same material system and both in field-free condition. Furthermore, switching direction of Type z and Type T modes is sensitive and invulnerable to external field, respectively. It means that one can physically define very different device properties just by controlling the direction of sourcing current relative to the easy axis of the in-plane layer, which is also benefit for practical applications. Besides, the stack used here is CoFeB/Ta/CoFeB system which is compatible with current MRAM industry and further lifts its impact for applications. The above results constitute main contributions of this study.

We also eagerly want to directly demonstrate realization of 4-state memory in this Type-T

switching mode, two states from switching of the perpendicular layer and the other two from switching of the in-plane layer. However, in current study, we can only use Hall signal to probe magnetization switching. 180° switching of the perpendicular layer can contribute significant anomalous Hall signals while 180° switching of the in-plane layer cannot physically contribute Hall signals in our devices due to twofold symmetry in angle dependence of planar Hall Effect (period of 180° in $\rho_{xy} \sim$ angle scan) [Tang et al. Phys. Rev. Lett. 90, 107201 (2003)]. Here we can only infer the switching of the in-plane layer by characteristics of the Type-T mode, without which the perpendicular layer has no means to be switched only by a current and an in-plane field normal to the current from symmetry-breaking viewpoint. This result is consistent well with static and dynamic simulations.

In order to directly measure the 4-state memory, we have already got down to optimize double-barrier full MTJ stacks with the i-CoFeB/Ta/p-CoFeB core structure, which would be undoubtedly a long-term project and could be reported in the subsequent manuscript but impossibly in this one. We sincerely pray for understanding from the referee on this point.

Main comments

Question 1. Hysteresis loops showing 4 different resistance states should be added in the main text. I suggest to discuss the impact of this result from a technological point of view.

Reply: We appreciate this insightful and helpful comments. As described above, 4 resistance states in such Hall devices are hard to measure because of incompetence of planar Hall Effect in monitoring 180° switching of the in-plane layer. In this manuscript, besides of Type z mode, we mainly accentuate that a novel symmetry-breaking law, relying on a switchable IMA layer, could enable new type of field-free switching (Type T mode), which is remarkably different from the previous studies reported by Miron [Nature 2011], Liu [PRL and Science, 2012], Fukami [Nature Mater. 2016] and Lau [Nature Nano. 2016]. However, the switching of the in-plane layer can be inferred by realization of SOT switching in Type T geometry: Without switching of the in-plane layer, a perpendicular layer has no means to be switched only by current and in-plane field normal to the current from viewpoint of symmetry breaking law (as commented by the 1st referee in his/her Question 1). We sincerely pray for understanding from the 2nd referee for the physical incompetence of the current stacks (without full MTJs) in directly measuring 4 state resistances.

Question 2. The hysteresis loop of the OPP field scan in Fig.2 a is not clearly visible.

Reply: We have added an inset of OOP hysteresis loop into the revised Figure. 2a for clearance.

Question 3. The resistance of the Ta should be estimated.

Reply: The effective resistivity of the stack is about $219 \mu\Omega \cdot \text{cm}$, close to the reported value ($\sim 191 \Omega \cdot \text{cm}$) in Ref. [Kim et al. Nature Mater. 12, 240 (2012)]. Here we have supposed current uniformly distributed in the three Ta layers and the two CoFeB layers in the Ta(5)/MgO(2)/CoFeB(1.0)/Ta(1.3) /CoFeB(1.4)/MgO(2)/Ta(3) layers. Here we have also corrected the stack thicknesses in the revised manuscript. The stack with ***T-type magnetic configuration*** is Ta(5)/MgO(2)/p-CoFeB(1.0)/Ta(1.3)/i-CoFeB(1.4)/MgO(2)/Ta(3) while the other

stack of Ta(5)/MgO(2)/p-CoFeB(1.2)/Ta(1.3)/p-CoFeB(1.2)/MgO(2)/Ta(3) has *two perpendicular CoFeB layers*. The former is used to realize Type T and Type z switching modes while the latter is used to characterize SOT efficiencies. We are extremely sorry for the trouble brought about by this mistake if any.

Question 4. The spin-Hall angle and the field like torque should be estimated.

Reply: We appreciate this insightful and helpful suggestion. To determine the field-like torque and damping-like torque here, we adopt a similar stack with a structure of Ta(5)/MgO(2)/FeCoB(1.2)/Ta(1.3)/FeCoB(1.2)/MgO(2)/Ta(3) where two perpendicular layers antiferromagnetically coupled via Ta layer of the same 1.3 nm thickness. It is phenomenally convenient to parameterize antidamping-like and field-like torques in terms of corresponding effective field H_{DL} and H_{FL} , using low frequency AC harmonic lock-in technique. We could use the following equations to calculate H_{DL} and H_{FL} by sweeping field in the x (parallel to current) and the y (transverse to current) direction, respectively.

$$\Delta H_{x(y)} = -2 \frac{\partial V_{2\omega}}{\partial H_{x(y)}} \bigg/ \frac{\partial^2 V_{\omega}}{\partial H_{x(y)}^2}$$

It is worthy of noting that the H_{DL} and H_{FL} of a T-type structure, one in-plane layer and the other perpendicular coupling layer, is not suitable to be directly measured by this method because the in-plane layer can be switched by relatively small in-plane field, complicating data interpretations.

Two lock-in amplifiers (Stanford - SR830) were employed to simultaneously measure the in-phase first harmonic (V_{ω}) and out-of-phase second harmonic ($V_{2\omega}$) signals. When applying an AC current in x direction along the channel, we can obtain effective H_{DL} (H_{FL}) by sweeping external field in x (y) direction. The dependence of the effective field on the channel current are show in Figure. **R4**. The longitudinal effective field ΔH_{DL} varies linearly with current density. The slope is about ± 0.6 mT/(MA/cm²). Sign depends on the remanence state (+z or -z direction), taking the form of $\Delta H_{DL} \propto \mathbf{m} \times \boldsymbol{\sigma}$ which is consistent with the theoretical analysis. In contrast, the signs of the slope of the transverse effective field ΔH_{FL} is ~ 0.744 mT/(MA/cm²) and falls into the form of $\Delta H_{FL} \propto \boldsymbol{\sigma}$, independent with the remanence direction. Overall, efficiency of the transverse effective field is ~ 0.24 times higher than that of longitudinal effective field, which is also consistent with the fact that field-like torque is remarkable in Ta/CoFeB systems [Zhang, XFH, et al PRB, 94, 174434 (2016).; Kim et al, Nature Mater, 12, 240-245 (2013)].

Figure R4 the effective damping-like field (a) and field-like field (b) as a function of current.

To estimate the spin Hall angle θ_{sh} (corresponding to anti-damping-like torque), we also employ an equation to calculate the effective θ_{sh} . The equation is as followed:

$$\theta_{sh} = \frac{2e}{\hbar} \frac{M_s t_F H_{DL}}{J}$$

Here e is the elementary charge, \hbar is the reduced Planck constant, θ_{sh} is the effective spin Hall angle, t_F is the absolute thickness of the ferromagnetic layer, respectively. J is the current density to measure H_{DL} . $M_s = -0.743$ MA/m is the magnetization of the perpendicular magnetized layer. θ_{sh} is derived to be ~ -0.167 , close to the reported values $-0.12 \sim -0.15$ in Liu's work [Science, 336, 555 (2012)].

Question 5. Which is the current density necessary for the switching at zero bias field? Compare this number with the other switching strategies at zero field.

Reply: The critical current density is about 18 and 33 MA/cm² for Type z and T mode, respectively, comparable with previous studies in Ta, Pt and W systems.

Reference	Material	J_c (A/cm ²)	θ_{sh}
Chen et al Appl. Phys. Lett. 111, 012402 (2017)	PtMn/CoFeB/Gd/CoFeB	$\sim 9.6 \times 10^6$ (Field Free, Type z)	0.084
Pai et al Appl. Phys. Lett. 101, 122404 (2012)	W/CoFeB/MgO	$\sim 1.8 \times 10^7$ (15mT, Type y)	-0.33
Liu et al PRL 109, 096602 (2012)	Pt/Co/MgO	$\sim 2.7 \times 10^7$ (10 mT, Type z)*	0.06
Fukami, et al. Nature Mater, 15, 535 (2016)	PtMn/[Co/Ni]	$\sim 1.8 \times 10^7$ (Field Free, Type z)*	0.1
Lau, et al. Nature Nanotech. 11, 758 (2016)	Pt/Co/Ru/CoFe/IrMn	$\sim 2.5 \times 10^7$ (Field Free, Type z)	0.124
Oh, et al. Nature Nanotech. 11, 878 (2016)	IrMn/CoFeB/MgO	$\sim 4.2 \times 10^7$ (Field Free, Type z)	Not found
Liu, et al. Science, 336, 555 (2012).	Ta/CoFeB/MgO	$\sim 1.5 \times 10^7$ (3.5mT, Type y)	-0.12~0.15
This work	CoFeB/Ta/CoFeB	1.8×10^7 (Field Free, Type z) 3.3×10^7 (Field, Free, Type T)	-0.167

*: if current density is not calculated by the authors, we calculate the current density supposing (1) the current uniformly distributes in the thickness direction and (2) shunting effect of high resistive underlayer are ignored.

Question 6. The size of the device is large for application in real magnetic devices, I believe that the switching process is not uniform. However, the macrospin model used for a simplified description of the device properties reproduce qualitatively the experimental behavior. Add a comment about this aspect.

Reply: SOT-driven domain wall motion requires two conditions: (1) DMI induced Néel-type domain wall and (2) effective perpendicular field experienced by the domain wall $\mathbf{H}_{\text{eff}} \propto \mathbf{m}_{\text{DW}} \times \boldsymbol{\sigma}$ as shown in Figure R5 [Pai et al. Phys. Rev. B 93, 144409 (2016)]. Here \mathbf{m}_{DW} is the direction of the center moment of a domain wall and $\boldsymbol{\sigma}$ is direction of spin polarization. In Type z mode, \mathbf{H}_x (parallel with I) aligns \mathbf{m}_{DW} in the x axis and $\boldsymbol{\sigma}$ is along the y axis. Thus \mathbf{H}_{eff} is along z axis (Figure R5).

However, in Type T mode, exchange coupling field experienced by the perpendicular layer is along the y axis (normal to I), which also inclines to align the \mathbf{m}_{DW} along the y axis and parallel to $\boldsymbol{\sigma}$. Thus \mathbf{H}_{eff} ($\propto \mathbf{m}_{\text{DW}} \times \boldsymbol{\sigma}$) becomes 0. The domain wall motion model failed to account for the behavior in Type T mode. Nevertheless we think SOT can still contribute to nucleation in incoherent switching process in Type T mode. The difference between this multidomain switching process and macrospin model is the anisotropic energy which should be replaced by an effective anisotropic energy of some local nucleation sites instead of the average one of the whole film. We have updated corresponding discussion in the revised manuscript.

Figure R5. Physical diagram for SOT-induced chiral domain wall motion from Ref. [Pai et al. Phys. Rev. B, 93, 144409 (2016)]. (a) Without and (b) with a field H_x along the x axis. Only with H_x , domains can grow up.

Question 7. Comment on the contribution to the

Reply: We sincerely apology for failing to reply this point. We have checked email and decision summary of Nature Communication system, finding no complete sentence here. We are pleased to further response to it later.

Question 8. Add a more detailed description of the contribution to the magneto resistance (MR). The IMA material should give an anisotropic MR while the PMA materials should contribute with the spin Hall resistance.

Reply: In this study we use Hall bar structure to pick Hall signals from the devices. In principle, the IMA and PMA layer can contribute a PHE (due to AMR or SMR) and AHE to the Hall signals, respectively. (Here we understand the “spin Hall resistance” as “anomalous Hall effect”.) However, as depicted in our Reply to Question 1 of the 2nd referee, planar Hall effect (due to AMR or SMR) would be twofold-symmetric in PHE vs angle scan [Tang et al. Phys. Rev. Lett. 90, 107201 (2003); Nakayama et al. Phys. Rev. Lett. 110, 206601 (2013)], which means 180° switching of the in-plane layer would not contribute to ΔR_{xy} and only 180° switching of the PMA layer can be detected in our Hall devices. We have revised the manuscript accordingly to explain this point.

Technical comment

More details about the spin-Hall geometry should be added, which material and how the current is injected in the stack.

Reply:

Etching details: When we pattern the device, three steps are implemented. First, useless area is etched out to SiO₂ substrate with the working ellipse area and the electrode areas retained. Second, the ellipse working area was covered by resist and the electrode areas were etched to the bottom MgO layer. In this case the IMA layer in the electrode areas were also all etched out. Third, Au electrodes were deposited to cover the etched electrode areas by sputtering and lift-off technology. Thus current can reach the CoFeB/Ta/CoFeB sandwich through side walls of a Hall device.

Measurement details: Anomalous Hall signals is measured by 4-terminal method. Device structure has shown in the manuscript. Then current from Keithley 2400 is injected along the blue arrows in Figure 2b, 3a and 4a. Voltage is measured along the direction transverse to current using Keithley 2182.

We have also revised method part accordingly.

Reviewers' comments:

Reviewer #2 (Remarks to the Author):

Second Report on the manuscript "Spin-orbit torque switching in a T-type magnetic configuration with current orthogonal to easy axes" by W.J. Kong et al.

I have carefully read the reply letter, and I believe that most of my concerns have been addressed.

I can understand that the measurements of the four resistance state is not possible in this device. I suggest to add in the conclusion some directions for the development of multistate resistance as also discussed in the rebuttal letter.

I absolutely agree with the authors "Nevertheless we think SOT can still contribute to nucleation in incoherent switching process in Type T mode.", I literature there are several papers concerning the non-uniform switching driven by the SOT, I suggest the authors to add one or two references for this in the main text.

In my opinion, the manuscript meets the criteria for Nat. Comm.

Reviewer #3 (Remarks to the Author):

In this manuscript, the authors reported a field-free switching of a perpendicular magnetized layer in a spin-valve structure induced by an in-plane electric current. The structure of the spin valve is CoFeB/Ta/CoFeB, where the first CoFeB has a perpendicular magnetic anisotropy (PMA) and the second CoFeB has an in-plane magnetic anisotropy (IMA). The so-called T-type configuration is different from what has been realized previously, such that the current is perpendicular, instead of parallel, to the magnetization of the IMA layer. While the result is interesting and surprising, I think there are significant flaws in the analysis.

The first major criticism is the symmetry analysis. In a Ta/CoFeB bilayer, an in-plane external magnetic field transverse to the current, which is very similar to the T-type configuration described in this paper, does not give rise to deterministic magnetization switching, due to the mirror symmetry of the system. However, the authors claim in the spin valve structure, the IMA magnetization reverses as current reverses, which necessarily violates the mirror symmetry. Therefore, deterministic switching is now allowed. I do not agree with this argument.

The left figure above is Fig. 3(a) in the manuscript that leads to the main finding. A current I_x can switch PMA magnetization from up to down, even when $H_x = 0$. The result is shown in the middle panel of Fig. 3(c). But if I rotate the figure by 180 degree (actually this is the same as if I change my viewing angle by tilting my head 180 degree), I get the right figure above. The sample and pattern should be symmetric under this operation. The only thing that's changed from the left figure is the current direction, and consequently the magnetization of IMA (this is still consistent with the theory in the paper). However, the PMA magnetization remains unchanged under this operation. This means under the same condition shown in the left figure, reversing the current will still lead to the same PMA magnetization switching: from up to down, which obviously contradicts the result shown in the paper.

Another way to put it, in the T-type configuration, what dictates the deterministic switching from up to down or vice versa? If there is no difference in energy or damping of the up or down state of the PMA, why an electric current would result in a preferred PMA magnetization?

As discussed above, deterministic switching is disallowed by symmetry. So, at best what the authors observe is some kind of toggle switch, i.e. regardless the initial state of PMA magnetization, when the current exceeds certain value, the magnetization will reverse. The toggle switch process does not violate symmetry, but it is of little interest to practical applications because the switching can be stochastic. One simple experiment to check whether

this is a toggle switch is to repeat the experiment in Figure 3 (c) with 0Oe external field. Reverse the current scanning, i.e. if initially the scanning was performed as +I -> -I -> +I, change it to -I -> +I -> -I, and see if the result is reversed. If it is the toggle switch, the result will be reversed. Conversely, if the result remains unchanged, it is likely to be a deterministic switching. However, even so, the apparent violation of symmetry discussed above should be addressed.

The second concern is that the authors claim the switching is correlated with the interlayer coupling (IC), and show in Fig. 4 that an external magnetic field H_y can offset the current-induced switching curve. But the magnitude of H_y seems too extreme. Even under the largest current applied, 40 mA, the effective field of the IMA cannot exceed 100 Oe (I estimate the largest possible Ampere's field to be 25 Oe). Therefore, under a $H_y = 1000$ Oe, the magnetization of the IMA cannot switch. If the IMA does not switch, what leads to the switching of the PMA? This seems to be inconsistent with the authors' argument.

In fact, I think the reason that H_y offsets the current-induced switching curve is not due to the field in the y-direction, but may be due to misalignment that results in a small field in the z-direction. Seen in the red curve in Fig. 4(b), even after the PMA magnetization is saturated in plane by a 6 kOe field, it still recovers 100% perpendicular magnetization with a preferred direction, when the in-plane field is removed. This strongly suggests a misalignment of H_y .

Given the inconsistency with symmetry, and some evidence that there could be potential misalignment of magnetic field, which may help break additional symmetry without being noticed, I suggest a major revision to address my concern, particularly the one with symmetry.

Responses to Reviewer Comments

We thank Reviewer #2 for insightful and positive comments and we have already added proper references to support in main text. Moreover, we sincerely appreciate Reviewer #3 for critical comments that have helped us to improve the clarity and quality of the revised manuscript. We provide point-to-point response to all comments by taking all comments into consideration and revised the manuscript accordingly.

Reviewer #2 (Remarks to the Author):

Second Report on the manuscript "Spin-orbit torque switching in a T-type magnetic configuration with current orthogonal to easy axes" by W.J. Kong et al.

I have carefully read the reply letter, and I believe that most of my concerns have been addressed.

I can understand that the measurements of the four resistance state is not possible in this device. I suggest to add in the conclusion some directions for the development of multistate resistance as also discussed in the rebuttal letter.

I absolutely agree with the authors "Nevertheless we think SOT can still contribute to nucleation in incoherent switching process in Type T mode.", I literature there are several papers concerning the non-uniform switching driven by the SOT, I suggest the authors to add one or two references for this in the main text.

In my opinion, the manuscript meets the criteria for Nat. Comm.

Reply: We are grateful for reviewer's positive and enlightening comments. We add three references to the revised main text concerning non-uniform switching, including Pai et al *Phys. Rev. B* **93**, 144409 (2016); Beach et al *Appl. Phys. Lett.* **104**, 092403 (2014) and Rojas-Sanchez et al *Appl. Phys. Lett.* **108**, 082406 (2016). The references are relevant with experimental observation of nucleation and non-uniform switching process, consisting with SOT mechanisms. We also add some perspective comments for the development of multi-state resistance in the conclusion part.

Reviewer #3 (Remarks to the Author)

In this manuscript, the authors reported a field-free switching of a perpendicular magnetized layer in a spin-valve structure induced by an in-plane electric current. The structure of the spin valve is CoFeB/Ta/CoFeB, where the first CoFeB has a perpendicular magnetic anisotropy (PMA) and the second CoFeB has an in-plane magnetic anisotropy (IMA). The so-called T-type configuration is different from what has been realized previously, such that the current is perpendicular, instead of parallel, to the magnetization of the IMA layer. While the result is interesting and surprising, I think there are significant flaws in the analysis.

Reply: We thank for the comments of Reviewer #3, which helped us to make a deep digging into the underlying physics of SOT driven magnetization switching in our CoFeB/Ta/CoFeB trilayer case. We will present more detailed static and dynamic discussion in this response letter. Furthermore, more new experimental results proposed by the referee have been conducted and added in response to some relevant comments.

The first major criticism is the symmetry analysis. In a Ta/CoFeB bilayer, an in-plane external magnetic field transverse to the current, which is very similar to the T-type configuration described in this paper, does not give rise to deterministic magnetization switching, due to the mirror symmetry of the system. However, the authors claim in the spin valve structure, the IMA magnetization reverses as current reverses, which necessarily violates the mirror symmetry. Therefore, deterministic switching is now allowed. I do not agree with this argument.

The left figure above is Fig. 3(a) in the manuscript that leads to the main finding. A

current I_x can switch PMA magnetization from up to down, even when $H_x = 0$. The result is shown in the middle panel of Fig. 3(c). But if I rotate the figure by 180 degree (actually this is the same as if I change my viewing angle by tilting my head 180 degree), I get the right figure above. The sample and pattern should be symmetric under this operation. The only thing that's changed from the left figure is the current direction, and consequently the magnetization of IMA (this is still consistent with the theory in the paper). However, the PMA magnetization remains unchanged under this operation. This means under the same condition shown in the left figure, reversing the current will still lead to the same PMA magnetization switching: from up to down, which obviously contradicts the result shown in the paper.

Another way to put it, in the T-type configuration, what dictates the deterministic switching from up to down or vice versa? If there is no difference in energy or damping of the up or down state of the PMA, why an electric current would result in a preferred PMA magnetization?

Reply: First, we would give a symmetrical analysis of Type-Z, Y and Type-T mode. We start with symmetry breaking law of Type-Z mode. As sketched in the Fig R1a. A perpendicular layer could be switchable by applying the charge current in the presence of external bias field along current. The switching polarity could be inferred by mirror operations. For example, if the experimental setup of area A is an equilibrium state (Fig R1a), after mirror symmetrical operation regarding XOZ (YOZ) plane, the setup in area C (B) will be another equilibrium state (considering magnetization and magnetic field both pseudo-vectors). So we can conclude that the fixed magnetic field along X axis breaks the XOZ plane mirror symmetry while the fixed current of X direction breaks that of YOZ plane.

For Type-Y mode, XOZ mirror operation produce no difference between area A and C as sketched in Fig R1b. Deterministic switching could only be obtained between area A and B. Opposite current direction breaks the YOZ mirror symmetry and realize switching between A and B states.

Type-T mode looks like a combination of Type Z and Y, as depicted in Fig R1c. YOZ mirror operation leads to the switch of both in-plane and perpendicular layers. It means, if area A is stable under positive current, area B will also be stable under negative current. Mirror symmetrical operation regarding XOZ plane leads to states in

A and C area. These two states have the same current direction and in-plane spins but opposite perpendicular spins. It means if the in-plane spin retains its direction, the current cannot lead to switching of the perpendicular spin between up and down states, because they are both stable. Reversely speaking, switching of the in-plane spin becomes a prerequisite condition to switch the perpendicular layer when there is no external field, which is the heart of Type-T mode. So we propose that a switchable IMA layer might break the XOZ mirror symmetry, leading deterministic switching of the PMA layer.

Fig R1, (a-c) The schematic mirror operation for Type-Z, Type-Y and Type-T modes, respectively. Light blue arrows indicate current direction. Red and dark blue arrows indicate direction of perpendicular and in-plane spins. Grey arrows indicate directions of applied fields.

Switching dynamical analysis, taking anisotropy, Zeeman splitting, interlayer coupling and damping-like torques into account, could provide detailed information. Results from the dynamical calculation is consistent with that from the symmetrical analysis. Noting that the parameters we take is the same as that of supplementary.

Switching dynamics of Mode Z and Y have been reported by Fukami et al. [Nat. Nano. 11, 621 (2016)] as shown in Fig R2e. In Type-Z mode, perpendicular spin will fast reverse to opposite direction while in Type-Y mode, spin experiences long precession near initial and final states, just like classic STT- mechanism.

In Type-T mode, current is applied in X direction and the EA axis of IMA layer is directed along Y. Fig R2 a-d shows field-free switching dynamics of the IMA and PMA layers when the current is above (below) J_{c2} , corresponding to Fig R2 b, d (a, c).

J_{c1} and J_{c2} are critical current density to switch IMA and PMA layers, respectively. As $J > J_{c2}$, m_{op-z} and m_{ip-y} both reverses their sign from an initial state, indicating their switching. It is worth noting that the IMA layer would also be switched when $J_{c1} < J < J_{c2}$, indicating the switching of IMA layer is a necessary but not sufficient condition to switch the PMA layer. Overall, the switching of the IMA layer provides us with a symmetry breaking element at field-free condition and leads to Type-T mode.

Fig R2. (a) and (c) the dynamic switching process of PMA and IMA layer, respectively, when current is below J_{c2} . (b) and (d) for that of current induced magnetization switching of PMA and IMA layer when $J > J_{c2}$. (e) Illustration of Type-z

and Type-y mode. [Fukami, et al. Nat. Nano. 11, 621 (2016).] (f) Mirror operation for current and current induced effective longitude and transverse field. [Yu et al. Nat. Nano. 9, 548 (2014)].

Regarding rotating device around z axis by 180° , though the comments of the 3rd referee is intuitively correct, we have to disagree with this symmetrical analysis method which is critical to distinguish final states of current-induced switching. The method suggested by the referee is actually 180° rotation while what we have adopted in the manuscript and in this reply is mirror symmetry. In the following part, we will give a typical example to identify the rationality of the above two methods (Figure R3).

Figure R3. Two symmetry analysis methods: (a) Rotation and (b) Mirror Symmetry. Red arrow indicates current direction. Blue arrows indicate directions of magnetization of the PMA and IMA layer.

Recently, several groups [Baek, et al. Nature Materials. <https://doi.org/10.1038/s41563-018-0041-5>; Wang, et al. Adv. Mater. DOI: 10.1002/adma.201801318] have studied a typical system with a PMA layer, an IMA layer and a spacer layer as shown in the left panel of Figure R3a. We suppose this state is the initial state as $I > 0$ (current flowing from left to right) without loss of generality. Then we analyze the final state as $I < 0$ (current from right to left) under the guide of the above two symmetry analysis methods. According to the 180° rotation symmetry around the z axis, as proposed by the referee, the final state would be as

shown in the right panel of Figure R3a. Instead, according to the mirror symmetry regarding YOZ plane, the final state would be as shown in the right panel of Figure R3b. According to the experimental results [Baek, et al. Nature Materials. <https://doi.org/10.1038/s41563-018-0041-5>; Wang, et al. Adv. Mater. DOI: 10.1002/adma.201801318], the observed final state satisfies the configuration of Figure R3b. Besides, the observed final state (IMA retaining its initial state but PMA layer being switched by current) could not be obtained by any rotation operations only. Thus, we incline that the mirror symmetry analysis may be more appropriate for our case as adopted by Yu et al. [Nature Nanotechnology, 9 (2014) 548–554] and Garello et al. [Nature Nanotechnology, 8, (2013) 587] in their works.

Besides of mirror symmetrical analysis in Figure R1 and R3, we also used a macrospin model to study the switching dynamic process in different measurement geometries. The macrospin model takes PMA, IMA, exchange coupling and damping-like spin-orbit torque into consideration. The model seems to qualitatively reproduce our observations as shown in the supplementary information. Besides, if we suppose current is applied along the easy axis of the IMA layer, the observations of Ref. [Baek, et al. Nature Materials. <https://doi.org/10.1038/s41563-018-0041-5>; Wang, et al. Adv. Mater. DOI: 10.1002/adma.201801318] could also be qualitatively obtainable by the model as shown in the supplementary of Ref. [Wang, et al. Adv. Mater. DOI: 10.1002/adma.201801318]. This coincidence indicates the reasonability of our model.

We have to admit the detailed switching mechanisms and even origins of spin-orbit torques in trilayer systems are still in hot debate [Taniguchi, et al. Phys. Rev. Appl. 3 (2015) 044001; Amin, et al. Phys. Rev. B 94 (2016) 104420; Freimuth, et al. arXiv:1711.06102v2] and need further study. The significance of our work lies mainly on experimentally realization of two different field-free switching modes in CoFeB/Ta/CoFeB system which has great application potential in MgO based SOT-MRAM devices. (One mode is sensitive to field while the other new one is not). Thus we weaken the argument on SOT mechanisms but instead stress the significance of the experimental realization of two-mode field-free switching in the revised manuscript.

As discussed above, deterministic switching is disallowed by symmetry. So, at best what the authors observe is some kind of toggle switch, i.e. regardless the initial state

of PMA magnetization, when the current exceeds certain value, the magnetization will reverse. The toggle switch process does not violate symmetry, but it is of little interest to practical applications because the switching can be stochastic. One simple experiment to check whether this is a toggle switch is to repeat the experiment in Figure 3 (c) with 0Oe external field. Reverse the current scanning, i.e. if initially the scanning was performed as +I \rightarrow -I \rightarrow +I, change it to -I \rightarrow +I \rightarrow -I, and see if the result is reversed. If it is the toggle switch, the result will be reversed. Conversely, if the result remains unchanged, it is likely to be a deterministic switching. However, even so, the apparent violation of symmetry discussed above should be addressed.

Reply: We have implemented the experiment proposed by the 3rd referee. At zero bias field, current induced magnetization switching is examined by opposite current switching orders +I \rightarrow -I \rightarrow +I (+-+ for short) and -I \rightarrow +I \rightarrow -I (-+- for short). The results are shown in Fig R4. Independent on current scanning orders, the switching polarity remains unchanged, indicating the switching is deterministic other than toggle switch process. It is notable that this switching experiment is reproduced for six times. The critical current shows small random fluctuations around 36 mA.

Fig R4. Six circles of magnetization switching with opposite current scanning orders (<+-+> and <-+->).

The second concern is that the authors claim the switching is correlated with the

interlayer coupling (IC), and show in Fig. 4 that an external magnetic field H_y can offset the current induced switching curve. But the magnitude of H_y seems too extreme. Even under the largest current applied, 40 mA, the effective field of the IMA cannot exceed 100 Oe (I estimate the largest possible Ampere's field to be 25 Oe). Therefore, under a $H_y = 1000$ Oe, the magnetization of the IMA cannot switch. If the IMA does not switch, what leads to the switching of the PMA? This seems to be inconsistent with the authors' argument.

In fact, I think the reason that H_y offsets the current-induced switching curve is not due to the field in the y -direction, but may be due to misalignment that results in a small field in the z direction. Seen in the red curve in Fig. 4(b), even after the PMA magnetization is saturated in plane by a 6 kOe field, it still recovers 100% perpendicular magnetization with a preferred direction, when the in-plane field is removed. This strongly suggests a misalignment of H_y .

Reply: We agree with the referee that there is possibly a small tilt from the Y axis toward the film normal as applying field nominally along the Y direction. This could lead to a small z -component field. First, we have to admit that it is nearly impossible to completely eliminate this mismatch. Second, we could prove that the small field in the Z direction (H_z) don't have a significant influence on the offset behavior, compared with so large a field in the Y direction.

We have committed supplementary experiment on a new device, which show same behaviors as before. As shown in Fig R5a, by controlling the angle between sample plane and H_y , we obtain H_y - R_{xy} loop at different angles. We find the 0° loop shows an important character: the equal R_{xy} value at +/- saturated field, indicating that the sample plane is closely aligned parallel with H_y . It is worth noting that the mismatch angle is less than 0.5° (the precision of the sample holder). ΔR_{xy} in respect with +/- saturated field due to mismatch increases when the tilting angle increases.

Figure R5b depicts the OOP as well as IP (0°) loops, which is similar with Figure 4b in the main text. In fact, before we measure current-induced switching curves, this angle calibration is always done as a precondition.

Moreover, Figure R5c show very similar transport properties as we showed in the

manuscript. While increasing H_y , the current-induced switching loops manifest a clear offset as well as shrinking. To verify the influence of small field along the Z direction, a simple experiment is to rotate sample at a fixed field.

Figure R5d shows the current-induced switching loops at different tilting angles ranging from 0° to 2° with a step of 0.5° . It is noticeable that a smaller angle than 0.5° would not generate observable difference, so we choose to rotate the sample with 0.5° step. Shown in Figure R5d, tilting the sample at a fixed H_y induced a couple of characteristics. (1) Negligibly small offset without significant shrinking is observed in the switching loops between 0° and 1.5° . (2) Magnetization would be fixed as the tilting angle reaches 2.0° . Further increase in H_y to 700 Oe will lead to disability of switching at 0.5° (Figure R5e), similar as that of 2.0° @500 Oe. Therefore the increase in H_y will narrow the tilting margin in which switching can be observed, confirming that the tilting angle in the main text is very small.

As we discussed above, the increase in H_y will eventually fix the magnetization and current switching is prohibited because of the small tilting angle. We observed the switching prohibition by increasing H_y to +/- 1500 Oe in the new sample in Figure R5f, while we still obtained switching loops at +/- 1500 Oe and fixed magnetization at +/- 2000 Oe in the main text. This may arise from anisotropy difference among devices. Overall, the above results imply that (1) the tilting angle in the main text is indeed small and (2) the small tilting angle, if exist, cannot explain observed offset of switching loops. Significant offset as well as shrink of switching loops is still probably owing to H_y itself.

Figure R5 (a) R_{xy} - H loops at different angles. (b) The in-plane and out of plane loops after calibration. (c) Current induced switching at different applied H_y when the sample is placed at 0° (d) Current induced switching when the applied field is fixed to be 500 Oe and changing tilting angle (e) No deterministic switching are observed when tilting the sample at 0.5° and 700 Oe H_y . (f) Large enough H_y will fix the magnetization at a certain up or down state.

Another important issue is the switching mechanism at very large H_y . We agree with the referee that the IMA layer could possibly not be switched in a large H_y . It is worth noting that the effective magnetic fields IMA and PMA layer are subjected to are smaller than applied H_y because of anti-parallelly exchanged coupling and dipolar interaction. The equilibrium state of IMA layer will title away from Y axis of XOY plane in the presence of effective longitudinal and transverse field from SOT. In such condition, a relatively large field could lower the switching barrier of the PMA as

reported by Miron et al [Nat. Mater. 9, 230 (2010)]. The field transverse to the current will help nucleation and cause partial switching with respect to certain charge current. In fact, for relatively large field H_y of 1500 Oe, results in our main text show that the switch starts at small current and the switching will be subjected to a multi-step switching process, indicating that the sample continues to switch through multidomain state in the present of large H_y in response to increased current, which is consistent with the discussion above. So the PMA layer is able to be still partially switched by applied current in a very large H_y .

Fig R6. The multidomain are generated by the combination of current and external field transverse to current channel. [Nat. Mater. 9, 230 (2010)]. H_{sd} is the effective transverse Rashba field.

Given the inconsistency with symmetry, and some evidence that there could be potential misalignment of magnetic field, which may help break additional symmetry without being noticed, I suggest a major revision to address my concern, particularly the one with symmetry.

Finally, we sincerely thank all the referees for their insightful and helpful comments because they help us a lot to modify and improve the quality of this manuscript. After a lengthy revision, we systematically addressed the SOT switching phenomenon in a T-type structure by experiment, static and dynamic analysis when the current is orthogonal to effective bias field. The simulation and experimental results qualitatively agrees with each other, unlocking a Type-T mode for SOT switching geometry.

In fact, after finishing field-free Type-Z switching in CoFeB/Ta/CoFeB multilayers by experiment and numerical simulation in the present of collinear current, we then

simulated the possibility of field free magnetization switching via an orthogonal current. The result was positive: the magnetization reversal seemed observed in a wide range of parameters. Then we carried out examination in such structures, consequently achieving field-free switching in Type-T mode. The most charming characteristic of Type-T mode is that the switching loop is not sensitive to external field (dramatically different from classic SOT-switching behaviors of PMA layer whose switching direction can be changed by field), implying its prospective applications in SOT-MRAM with high stability and robustness against external disturbance. Overall speaking, two switching modes (Type Z and Type T) characterized by different switching behaviors and experimentally realized in the same MRAM-compatible structure (CoFeB/Ta/CoFeB) would at least provide an opportunity for community to design more functional SOT devices such as multi-state memories or programmable logic devices via different switching modes.

Your reconsideration and enlightening comments are always highly appreciated by the authors.

Reviewers' comments:

Reviewer #3 (Remarks to the Author):

In the revised version, the authors have made more arguments about the symmetry analysis and carried out a measurement that I proposed. However, I am still not convinced:

1. First of all, it is difficult to argue a spintronics experiment is wrong simply based on theoretical argument, because there could be many variations in the experiment that may not have been taken into account by theory. However, symmetry analysis is an exception. A phenomenon is real only if it is allowed by symmetry. If symmetry forbids a phenomenon that is still observed in an experiment, there must be hidden symmetry-breaking that was not noticed in the experiment. The symmetry here includes all symmetries of the point group (e.g. inversion symmetry, rotational symmetry), not just mirror symmetry.
2. But even if we only consider the mirror symmetry, I still disagree with the author's analysis. Below is Fig. R1(c) in the response letter. If the switching case in area A is possible, the switching case in area C must be true, where the mirror symmetry is applied to the entire system. The fact that the switching cases in A and C contradict each other, actually indicate that neither A or C can be true.

3. The authors suggest the mirror symmetry operation between case A and case B are consistent with each other in the above figure. Let's say cases A and B are both valid. Try to look at case B from the opposite direction, i.e. from $-y$ direction. What you see is case A except the perpendicular magnetization is reversed. Again, this contradiction suggests case A cannot be a deterministic switching.
4. To sum up 2 and 3, here is a challenge: In case A, let's say a current in the $-x$ direction can switch PMA from down to up ($-z$ to z). (a) Rotate the whole device together with the current leads about the z -axis, I suppose the PMA magnetization should still switch from down to up. (b) Take the same device, disconnect the current leads, rotate the device 180 degree about the z -axis, and then reconnect the current leads. How to apply a current to realize the same PMA magnetization reversal? Do you see conflicting results between (a) and (b)?
5. The authors cited Baek and Wang's papers as a counter argument toward the rotational symmetry. However, the switching mechanisms in both papers, where current is parallel with IMA magnetization, are actually allowed by both mirror symmetry and rotational symmetry. Therefore, I do not understand the rationale of choosing mirror symmetry over rotational symmetry when analyzing the configuration of this paper. Nevertheless, the symmetry argument above (2, 3 and 4) needs to be addressed.

The authors provide a simulation that suggests a deterministic switching in this trilayer system in Fig. S2 and Fig. R2 (a-d). This can be very helpful for understanding the switching mechanism, because in simulation it is relatively easy to eliminate any hidden broken symmetry. Because of the interlayer exchange coupling, the magnetization in both PMA and IMA are tilted in the initial condition, as illustrated in case A below. When I apply mirror symmetry to case A, I get case B, which means the two configurations are equally unstable or equally stable. Therefore, I don't think one can realize a deterministic switching. Below is a possible simulation to check.

For the simulation in Fig. R2(b,d), the present initial condition is $\mathbf{m}_{\text{PMA}} = (0,0,1)$ and $\mathbf{m}_{\text{IMA}} = (0,-0.6,-0.1)$. This initial condition is a bit odd. I expect both magnetizations to tilt, \mathbf{m}_{IMA} to have a normalized total magnetization and the two magnetizations should be at equilibrium before the application of a current pulse. Nevertheless, my prediction is that if the initial condition is changed to $\mathbf{m}_{\text{PMA}} = (0,0,-1)$ and $\mathbf{m}_{\text{IMA}} = (0,-0.6,0.1)$ (Note not reversing the whole magnetization, but only reversing the magnetization in the z-direction), the PMA magnetization should be switched from down to up, due to the mirror symmetry. Can the authors use their macrospin model to confirm or disapprove this?

Symmetry remains my biggest concern about this paper. I recommend the authors to convince me on the issues I raised above.

Reply to the 3rd referee

First of all, we would like to sincerely appreciate the 3rd referee for his/her insightful comments regarding symmetry analysis, which has really inspired us to implement deep thinking and analysis based on detailed spin dynamic simulation as shown below. The simulations have unambiguously confirmed correctness of the symmetry analysis put forward by the 3rd referee and shed lights on a hidden but critical symmetry-breaking factor: easy axis (EA) tilting of PMA layer which could be evidenced among our experiments. According to the suggestion of the referee, we have thoroughly revised the manuscript and corresponding supplementary information accordingly. In order to save time of the referee, we have also introduced the model and all additional information in this reply. Therefore, **reading through this reply or supplementary information would be enough to understand the model.**

This reply includes the following contents: first part is our point-to-point brief answer to the comments or questions of the referee. Second part is a detailed introduction to the revised model which includes **(1)** a revised symmetry-breaking analysis (Figure R1), **(2)** the reason why switching M_z of PMA layer in the original system is impossible (Figure R2), **(3)** the detailed model of Type-z switching in a PMA-IMA coupled system (Figure R3-R6) and **(4)** the detailed model of Type- T switching in the coupled system (Figure R7-R13).

1. Brief Point-to-Point Reply as well as Comments from Referee

In the revised version, the authors have made more arguments about the symmetry analysis and carried out a measurement that I proposed. However, I am still not convinced:

(1.1) First of all, it is difficult to argue a spintronics experiment is wrong simply based on theoretical argument, because there could be many variations in the experiment that may not have been taken into account by theory. However, symmetry analysis is an exception. A phenomenon is real only if it is allowed by symmetry. If symmetry forbids a phenomenon that is still observed in an experiment, there must be hidden symmetry breaking that was not noticed in the experiment. The symmetry here includes all symmetries of the point group (e.g. inversion symmetry, rotational symmetry), not just mirror symmetry.

Reply: We agree with the referee on this point. The deterministic switching is indeed forbidden in our previous model. The model cannot break 180° rotational symmetry and xoz mirror symmetry [also See 2. (1) in the following detailed reply]. Thus only IMA layer can be switched while PMA layer cannot in the previous model [also See 2. (2) in following the detailed reply].

(1.2) But even if we only consider the mirror symmetry, I still disagree with the author's analysis. Below is Fig. R1(c) in the response letter. If the switching case in area A is possible, the switching case in area C must be true, where the mirror symmetry is applied to the entire system. The fact that the switching cases in A and C contradict each other, actually indicate that neither A or C can be true.

Figure C1 in the referee's comment

Reply: We agree on the fact that in the above figure State in A and C region is symmetrical regarding xoz plane, which means the same current cannot determine PMA layer spin-down or up. A hidden symmetry breaking factor has to be added into system to realize PMA layer switching. As shown in 2. (1) in the following detailed reply, a tilting EA of PMA layer toward EA of IMA layer can be this key factor. Besides, time-resolved spin dynamic simulation [2. (4) in the following detailed reply] proves the role of the EA tilting. The tilting can be experimentally evidenced by several observations such as (a) significant offset in R_{xy} vs I in Type-z mode and (b) only one specific chirality in R_{xy} vs I in Type-T mode being obtained. These observations can be reproduced after the EA tilting is taken account into the original model.

(1.3) The authors suggest the mirror symmetry operation between case A and case B are consistent with each other in the above figure. Let's say cases A and B are both valid. Try to look at case B from the opposite direction, i.e. from $-y$ direction. What you see is case A except the perpendicular magnetization is reversed. Again, this contradiction suggests case A cannot be a deterministic switching.

Reply: We agree on this point. Looking at Case B from the $-y$ direction leads to a state similar with A except the opposite spin of PMA layer, which indicates B cannot be switched to A in the original model. We have introduced the reason from spin dynamic analysis in 2. (2) of the following detailed reply]. However, this symmetry, 180° rotation, can also be broken by the EA tilting proposed above as shown in 2. (1) and 2. (4) of the following detailed reply.

(1.4) To sum up 2 and 3, here is a challenge: In case A, let's say a current in the $-x$ direction can switch PMA from down to up ($-z$ to z). (a) Rotate the whole device together with the current leads about the z-axis, I suppose the PMA magnetization should still switch from down to up. (b) Take the same device, disconnect the current leads, rotate the device 180 degree about the z-axis, and then reconnect the current leads. How to apply a current to realize the same PMA magnetization reversal? Do you see conflicting results between (a) and (b)?

Reply: We have realized that, in the original model, current can **only switch IMA layer** but **cannot switch PMA layer**. Spin dynamic analysis in 2. (2) of this reply can prove this point. In order to realize deterministic switching of the PMA layer, extra symmetry-breaking factor has to be taken into account. In the following detailed reply, we will prove an EA titling of PMA layer toward proper direction can achieve the switching in Type-T mode [2. (4) of this reply]. The tilting is also evidenced by several experimental

observations.

(1.5) The authors cited Baek and Wang's papers as a counter argument toward the rotational symmetry. However, the switching mechanisms in both papers, where current is parallel with IMA magnetization, are actually allowed by both mirror symmetry and rotational symmetry. Therefore, I do not understand the rationale of choosing mirror symmetry over rotational symmetry when analyzing the configuration of this paper. Nevertheless, the symmetry argument above (2, 3 and 4) needs to be addressed. The authors provide a simulation that suggests a deterministic switching in this trilayer system in Fig. S2 and Fig. R2 (a-d). This can be very helpful for understanding the switching mechanism, because in simulation it is relatively easy to eliminate any hidden broken symmetry. Because of the interlayer exchange coupling, the magnetization in both PMA and IMA are tilted in the initial condition, as illustrated in case A below.

Figure C2 from the referee's comment

When I apply mirror symmetry to case A, I get case B, which means the two configurations are equally unstable or equally stable. Therefore, I don't think one can realize a deterministic switching. Below is a possible simulation to check.

For the simulation in Fig. R2(b,d), the present initial condition is $m_{\text{PMA}} = (0,0,1)$ and $m_{\text{IMA}} = (0,-0.6,-0.1)$. This initial condition is a bit odd. I expect both magnetizations to tilt, m_{IMA} to have a normalized total magnetization and the two magnetizations should be at equilibrium before the application of a current pulse. Nevertheless, my prediction is that if the initial condition is changed to $m_{\text{PMA}} = (0,0,-1)$ and $m_{\text{IMA}} = (0,-0.6,0.1)$ (Note not reversing the whole magnetization, but only reversing the magnetization in the z-direction), the PMA magnetization should be switched from down to up, due to the mirror symmetry. Can the authors use their macrospin model to confirm or disapprove this? Symmetry remains my biggest concern about this paper. I recommend the authors to convince me on the issues I raised above.

Reply: In the following simulation work, we have adopted the following procedure as suggested by the referee. First calculate the equilibrium or quasi-equilibrium states of the system without current and then obtain time-evolution of the system at elevating current.

Spin dynamic simulation results to the question proposed by the referee is as following.

In the original model without EA tilting, if system starts from State A (B) at $I=0$, it can only go to State D (C) after a negative current pulse. Definition of State A-D is as following. The results show only IMA layer can be switched while PMA layer cannot as also shown in 2.(2) of the following detailed reply.

In the revised model with EA tilting, system will finally arrive at State B' or D' as following after a large enough positive or negative current pulse as also shown in 2.(4) of the following detailed reply.

2. Detailed Reply Part

(1) SOT-symmetry-breaking in PMA-IMA coupled system

Figure R1 shows xoz and yoz mirror symmetries of a system with a PAM layer, an IMA layer and their interlayer coupling. Region I and III indicate opposite currents possibly lead to opposite IMA and PMA layers. However, this switching becomes possible only if xoz mirror symmetry is broken, because Region I and II indicate opposite PMA layers but under the same current. If EA of the PMA layer tilts toward EA of the IMA layer as shown by a red line in Region I of Figure R1, this xoz mirror symmetry can be broken. The xoz mirror symmetry results in opposite EA' tilting (a dashed line in Region II), which is impossible for a uniaxial PMA layer with a single and fixed EA. This EA tilting might result from the multi-step annealing process in experiment. In the 3rd part, we will show this EA tilting can be evidenced from Type-z switching. In the 4th part, we will show this EA tilting is necessary to realize Type-T mode. Worth noting, this EA tilting can also break 180° rotational symmetry raised by the referee.

Figure R1. Symmetry breaking law of Type-T mode. Green, blue and red arrows denote current, IMA layer and PMA layer, respectively. A red solid and dashed line in Region I and II indicate tilted EA of PMA layer.

(2) Spin dynamic analysis for the original system without EA tilting

We then applied time-dependent spin-dynamics to study switching behavior of system

in Type- T mode, which becomes indeed a powerful tool to verify the results from symmetry analysis. The simulation details can also be found in the revised supplementary information.

Considering a system with a PMA layer without EA tilting, an IMA layer (with EA along the y axis) and antiferromagnetic coupling between them, we can easily obtain 4 stable states named as A, B, C and D as shown in the insets of Figure R2. If we further apply an orthogonal current I_x to activate the system originally at the initial states, we can obtain diverse final states within different torque ranges as shown in the main panels of Figure R2.

Figure R2. Final states evolved from the 4 different initial states whose initial configurations are shown by the corresponding insets.

Figure R2a shows system has to pass through State D if it is initialized from State A and destined to State C with negatively increasing torque. However, once the system goes into State D, we have to adopt Figure R2d to further analyze switching order since the initial state has been changed to State D. Figure R2d shows that the system will persistently remain in State D with further negatively increasing torque. This means the transition $A \leftrightarrow C$ is actually forbidden while the transition $A \leftrightarrow D$ is permitted. Similar arguments based on Figure R2b and c show the transition $B \leftrightarrow D$ is forbidden while the

transition $B \leftrightarrow C$ is permitted. Thus our time-dependent spin-dynamic results clearly show the picture that large enough I_x in Type- T mode can lead to switch of the IMA layer *only* with the PMA layer retaining its perpendicular orientation during the whole process.

However, our experimental observation unambiguously shows a distinct scheme of switching PMA layer with fixed switching direction and insensitivity to external fields, compared with classic Type- z mode. It means that there should be some other factor(s) which we have ignored in previous analysis to break rotational symmetry as innovatively proposed by the referee. In the following, we will show that a small tilt of EA of the PMA layer toward $-y$ axis can qualitatively reproduce the observed phenomena including switching behaviors in Type- T mode as well as abnormality in Type- z mode.

(3) Type- z mode

We first analyze Type- z mode: EA of the IMA layer and current I_x are both along the x axis. Experiments show two features: (1) opposite magnetizing history of the IMA layer reverses switching direction of the PMA layer and (2) there is remarkably negative offset in R_{xy} vs. I_x hysteresis loops. The following model with an assumption of tilt EA of the PMA layer toward the x axis (EA of IMA layer) can account for both features. System energy E_T is composed by effective anisotropies of both layers, antiferromagnetic coupling and Zeeman splitting.

$$E_T = E_{OP} + E_{IP} + E_{IEC} + E_Z \quad (\text{R1})$$

$$E_{OP} = -K_1 \cos^2 \beta$$

$$E_{IP} = -K_2 \sin^2 \theta_2 [\eta + (1 - \eta) \sin^2 \phi_2]$$

$$E_{IEC} = A [\sin \theta_1 \sin \theta_2 \cos(\phi_1 - \phi_2) + \cos \theta_1 \cos \theta_2]$$

$$E_Z = -HM_1 \sin \theta_1 \cos(\phi_1 - \phi_H) - HM_2 \sin \theta_2 \cos(\phi_2 - \phi_H)$$

Here θ and ϕ is polar and azimuth angle, respectively. $\theta=0$ for $+z$ axis and ($\theta=90^\circ$, $\phi=0$) for $+x$ axis. EA and magnetization of the PMA layer is $(\sin\theta_0\cos\phi_0, \sin\theta_0\sin\phi_0, \cos\theta_0)$

and $(\sin\theta_1\cos\phi_1, \sin\theta_1\sin\phi_1, \cos\theta_1)$, respectively. Angle β between the EA and the PMA layer is thus defined by $\cos\beta = \sin\theta_1\sin\theta_0\cos(\phi_1 - \phi_0) + \cos\theta_1\cos\theta_0$. Magnetization of the IMA layer is $(\sin\theta_2\cos\phi_2, \sin\theta_2\sin\phi_2, \cos\theta_2)$. K_1 and K_2 are effective PMA and IMA energies, respectively. Parameter $1-\eta$ denotes the ratio of in-plane uniaxial anisotropy to total in-plane anisotropy. A is antiferromagnetic coupling constant. H is applied field. M_1 and M_2 are saturated magnetization of the PMA and IMA layers, respectively. ϕ_H is azimuth angle of external field. $\phi_H=0$ for $+x$ axis and $\phi_H=90^\circ$ for $+y$ axis.

I_x induces spin current σ with polarization along the y axis via spin Hall Effect. Then spin currents are transferred up and down to both ferromagnetic layers and activate spin dynamics and switching. The two ferromagnetic layers sandwich the Ta layer. Thus spin currents absorbed by them have opposite signs. Spin current σ has dimension of K_1/M_1 in this model. As provided spin currents are not too high, equilibrium magnetizations of both layer are constrained in XOZ plane. In this case, the final state of the system can be obtained via torque equilibrium condition Equation (R2).

$$\begin{aligned} K_1 \sin 2(\theta_1 - \theta_0) - A \sin(\theta_1 - \theta_2) - HM_1 \cos \theta_1 - a_1 \sigma &= 0 \\ K_2 \sin 2\theta_2 - A \sin(\theta_1 - \theta_2) + HM_2 \cos \theta_2 - a_2 \sigma &= 0 \end{aligned} \quad (R2)$$

The parameter $a_{1/2}$ characterizes interfacial absorption efficiency of spin current for the PMA/IMA layer. Further from Equation R1, it is straightforward to obtain an Eigen Equation (R3) which determines system stability and evolution routes.

$$\begin{bmatrix} d\theta_1 \\ d\theta_2 \end{bmatrix} = \Pi^{-1} \begin{bmatrix} a_1 \\ a_2 \end{bmatrix} d\sigma \quad (R3)$$

$$|\Pi| = [2K_1 \cos 2(\theta_1 - \theta_0) + HM_1 \sin \theta_1 - A \cos(\theta_1 - \theta_2)][2K_2 \cos 2\theta_2 - HM_2 \sin \theta_2 + A \cos(\theta_1 - \theta_2)] + A^2 \cos^2(\theta_1 - \theta_2)$$

$$\Pi^{-1} = \frac{1}{|\Pi|} \begin{bmatrix} [2K_2 \cos 2\theta_2 + A \cos(\theta_1 - \theta_2) - HM_2 \sin \theta_2] & -A \cos(\theta_1 - \theta_2) \\ A \cos(\theta_1 - \theta_2) & [2K_1 \cos 2(\theta_1 - \theta_0) - A \cos(\theta_1 - \theta_2) + HM_1 \sin \theta_1] \end{bmatrix}$$

System is stable as $|\Pi| < 0$ and becomes unstable as $|\Pi|$ approaching 0. We are especially interested in the case of zero external field. In order to derive switching behaviors, we have to first obtain initial states at $\sigma=0$ and then apply Equation (R3) to move system

toward final states at σ with iterative algorithm.

There are 4 initial states available at zero current. They are named as A, B, C and D as shown in Figure R3. We have used the following set of parameters here as well as in the coming calculations. $K_1=1$, $K_2=2$, $A=0.6$, $\eta=0.5$, $M_1=1$; $M_2=0.6$, $\theta_0=-10^\circ$, $a_1=1$; $a_2=0.6$.

Figure R3. Available initial states at zero current with the above set of parameters.

The σ dependence of determinant of the Eigen matrix $|\Pi|$ is shown in Figure R4. Two features are worth highlighting. First, States A and C (States B and D) share the same region where both of them are stable. Out of the region they become unstable simultaneously. It means transitions $A \leftrightarrow C$ and $B \leftrightarrow D$ are forbidden. Second, critical σ to induce transitions at positive and negative directions are distinct ($\sigma_1 \neq \sigma_2$), indicating offset M vs. σ curves.

In order to further look into switching route, we have inserted critical torque σ_c as well as spin configurations $(\theta_{1c}, \theta_{2c})$ at the corresponding transition point into Equation (R4) which is integrated from Eq. (R2) with $H=0$. Parameters $\varepsilon_{1/2}$ with energy dimension are mathematically constructed as Equation (R4) to help stability analysis at transition points. Minimum locations in ε_i vs. θ_i curves indicate steady state at a transition point. The ε_i vs. θ_i curves in Figure R5 can thus tell switching route.

$$\begin{aligned}\varepsilon_1 &= -\frac{1}{2}K_1 \cos 2(\theta_1 - \theta_0) + A \cos(\theta_1 - \theta_{2c}) - a_1 \sigma_c \theta_1 \\ \varepsilon_2 &= \frac{1}{2}K_2 \cos 2\theta_2 + A \cos(\theta_{1c} - \theta_2) + a_2 \sigma_c \theta_2\end{aligned}\tag{R4}$$

Figure R4. Torque dependence of $|\Pi|$ with different initial states.

Figure R5 Stability analysis of State A, B, C and D at $\sigma=0.6$

For example, State A turns unstable at $\sigma=0.6$. We can see from Figure R5a that instability of system is only brought about by the PMA layer (M_1) whose stable positions are indicated by blue dots. Figure R5c-d shows very different stable positions

(green dots) of M_2 with Figure R5a-b, indicating neither $A \leftrightarrow C$ nor $A \leftrightarrow D$ transitions in this case are feasible because the IMA layer is in stable position and cannot switch. In fact, Figure R5a-b shows M_1 can naturally slip from State A into State B at $\sigma_1=0.6$. According to the above analysis, we can conclude transition $A \leftrightarrow B$ is the only permitted switching route for the system as State A becomes unstable. Similar case occurs for the transition $C \leftrightarrow D$.

Therefore the final switching diagram in Type-z mode can be depicted by Figure R6. There are three noticeable characteristics in the diagram. (1) M_{1z} switches while M_{2x} retains its orientation during transitions. (2) Switching direction of M_{1z} , clockwise or counterclockwise, depends on direction of M_{2x} , along the $-x$ or $+x$ axis, respectively. (3) There are apparently the same offsets of M_{1z} vs. σ hysteresis loops in both Transitions $A \leftrightarrow B$ and $C \leftrightarrow D$. Offset direction depends on tilting angle of EA of the PMA layer. For example, negative offset results from negative tilt angle. Characteristics (1) and (2) have been reported in similar structures by Wang et al [*Adv. Mater.* **30** (2018) 1801318] and Baek et al [*Nat. Mater.* **17** (2018) 509]. The 3rd characteristic is unique here, which can be caused by tilt of EA of the PMA layer toward proper direction. No offset is predicted if the tilt is absent.

Figure R6. M vs. σ switching loops in Type- z mode of PMA-IMA coupled system with EA of the PMA layer tilt toward the $-x$ axis by a small angle. The parameters in Figure R3 are adopted here.

(4) Type- T mode

Significance of this paper is contributing to spin-orbitronics a new switching scheme of PMA materials with robust external-field immunity. Distinguished from classic Type- z mode discussed above, in-plane current is applied perpendicularly to EA of the IMA layer in this new scheme. In the following, we will prove the scheme is physically feasible and the deterministic switching can be reproduced if EA of the PMA layer tilts properly. Coordinate system in Type- T mode is different from that in Type- z mode: current is applied along the x axis while EA of the IMA layer is aligned along the y axis. In this case, EA of the PMA layer should also tilt toward the y axis (also the EA of IMA layer here). The same parameters in Figure R3 are adopted. We can also obtain 4 states at $I_x=0$ in Figure R7, which are virtually the same with those in Figure R3.

Figure R7. Initial states available in Type- T mode as $I_x=0$ and $H=0$. They are essentially the same with those in Figure R3. Only difference is that the x axis in Figure R3 is replaced by the y axis in Figure R7 due to reset of measurement setups in Type- T mode.

To analyze Type- T mode, we have to adopt time-dependent LLGS Equation (R5).

$$\frac{\partial \vec{m}_i}{\partial t} = -\gamma \vec{m}_i \times \vec{H}_{i,\text{eff}} + (-1)^{i+1} a_i \vec{m}_i \times \vec{\sigma} \times \vec{m}_i + \alpha \vec{m}_i \times \frac{\partial \vec{m}_i}{\partial t} \quad (\text{R5})$$

Here $\vec{H}_{i,\text{eff}}$ is effective field experienced by the i^{th} layer, 1st and 2nd layer for PMA and IMA layer, respectively. $\vec{H}_{i,\text{eff}}$ can thus be obtained from Equation (R1). The parameter α is damping constant and set as 0.1. The parameter a_i characterizes interfacial absorption efficiency of spin current for the i^{th} layer. After I_x is injected or polarization of spin current $\vec{\sigma}$ with dimension of K_1/M_1 polarizes along the y axis, spin dynamics of the system is activated. The final steady states started from different initial states under variable σ are summarized in Figure R8.

Figure R8. Final steady states (denoted by red characters as insets) started from the 4 initial states (marked by black characters on top) driven by SOT (denoted by σ). Dotted lines indicate transition from one state to another. After applying large enough torque, only transition $B \leftrightarrow D$ becomes visible or in other words States B and D are dynamically reachable even though system initially locates at State A or C.

Figure R8 tells that system will be finally stabilized into State D and B, respectively, at large enough negative and positive torques even though it initially locates at State A or

C. State A and C turn unstable at large enough positive and negative torques. Therefore, the transition $B \leftrightarrow D$ will become the only permitted switching route driven by torque in this case as shown by Figure R9. The following points with comparison with Type-z mode are worthy of special attentions. (1) Type- T mode switches both IMA and PMA layers **simultaneously** while Type-z mode switches PMA layer only. (2) Switching direction (clockwise) in Figure R9a is determined by EA tilting angle of the PMA layer. Here we have used $\theta_0 = -10^\circ$. If we use $\theta_0 = +10^\circ$, switching direction of the M_{1z} vs. torque loop is also changed to counterclockwise (Figure R10a). In this sense, there should be a fixed relation between switching direction in Figure R9a (or Figure 3c) and offset direction in Figure R6a,c (or Figure 2c,d).

Figure R9. Switching curves of M_{1z} and M_{2y} driven by spin-orbit torque in Type- T mode for the transition $B \leftrightarrow D$. Here $\theta_0 = -10^\circ$.

Figure R10. Switching curves of M_{1z} and M_{2y} driven by spin-orbit torque in Type- T mode for the transition $A' \leftrightarrow C'$. Here $\theta_0 = +10^\circ$. State A' and C' is slightly changed from State A and C due to the change in θ_0 .

It will be helpful to switch some parameters such as A or EA tilting angle θ_0 off in the model to investigate their roles in Type- T switching with the other parameters unchanged. If $\theta_0 = 0^\circ$, switching behaviors have already been presented in Figure R2. Only the IMA layer is switchable by I_x while M_z component of the PMA layer cannot be switched. If $A=0$, the IMA is still switchable with a larger critical switching current than in Type- T mode. More importantly, the PMA layer can be only rotated in-plane instead of 180° deterministic switching as shown by Figure R11a,c. Therefore interlayer coupling and easy axis tilting of PMA layer are both indispensable in this switching scheme.

Figure R11. Switching curves as $A=0$ and $\theta_0 = -10^\circ$ with the other parameters same with those in Figure R3. Only IMA layer is switchable while the PMA layer can only be rotated in-plane. Here State A'' - D'' are slightly deviated from State A - D due to zero A . Normalized switching orbitals as **c**, $A=0$ and $\sigma=0.75$ and **d**, $A=0.6$ and $\sigma=0.5$. Spin

dynamics are initialized from State D'' in **c** or State D in **d**. If $A=0$, magnetization of the PMA layer can only reach an in-plane state as shown in **c**. If $A>0$, the PMA switches its M_z component at the same time when the IMA switches its M_y component.

Spin dynamics with $A=0$ and $A=0.6$ shown in Figure R11**c,d** also indicates the switching mechanism of Type- T mode. System initially locates at State D'' or D. In a system without coupling ($A=0$), a large I_x (or $+\sigma_y$) can drive magnetization of the PMA layer from Point 1 to an in-plane position around Point 2. In this case, \mathbf{M}_1 is nearly parallel with $+\sigma_y$. Instead, the IMA layer absorbs spin current with opposite direction ($-\sigma_y$), which is thus switched from Point 2 to Point 4. If the coupling is switched on ($A=0.6$), \mathbf{M}_1 is still first tilt to a location near Point 2. Owing to antiferromagnetic coupling, system becomes very unstable when \mathbf{M}_1 and \mathbf{M}_2 both parallelly locate near Point 2. Thus once \mathbf{M}_2 switches, system relaxes to its lowest energetic state, State B with M_{1z} pointing down. It is the reason why M_{1z} and M_{2y} switch simultaneously in this mode.

Immunity of Type- T mode to external fields

We have also experimentally studied switching behaviors of Type- T mode under different $H_{x/y}$. The results in Figure 3-4 show switching direction of Type- T mode cannot be changed by the external fields, which is remarkably different from the behavior of Type- z mode. Furthermore, H_y can offset M_{1z} vs. torque (denoted by σ) loops. These features can both be reproduced by the above time-dependent LLGS model as shown in Figure R12-R13.

Especially, as shown in Figure R8, M_{1z} switching is highly correlated with M_{2y} switching. **They switch at the same critical torques.** Now an external H_y is applied along the easy axis of the in-plane layer. On the basis of a uniaxial anisotropy, the in-plane layer obtains additional unidirectional anisotropy. This unidirectional anisotropy will make M_{2y} switching in one direction easier while in the opposite direction harder, thus offsetting the corresponding M_{2y} vs σ curves. Since M_{1z} switching depends on the M_{2y} switching, an offset would also occur in the corresponding M_{1z} vs σ curves as shown in Figure R13 which qualitatively reproduces the feature of Figure 4 of the main text.

Figure R12. Calculated M_{1z} vs. torque hysteresis loops under different H_x . Switching direction cannot be changed by H_x .

Figure R13. Calculated M_{1z} vs. torque hysteresis loops under different H_y . H_y can offset the loops by affecting switching of IMA layer which is a preliminary condition to switch PMA layer since the two layers switch simultaneously at the same critical torque as shown in Figure R8.

Comparison between Type-T mode and Type-z mode

We give a thorough comparison between Type-T mode and Type z mode in Table R1 as a summary.

Table R1. Comparison between Type- T mode and Type z mode based on the above calculations with the parameters in Figure R3.

Field	Type- T mode	Type- z mode
Switching layers	IMA and PMA layers switch at the same torque	Only PMA
Material requirement	$A \neq 0$, EA tilting of PMA layer toward EA of IMA layer	$A \neq 0$
Results of EA tilt of PMA layer	Indispensable for Type- T mode	Offset M_{1z} vs. torque loops
Switching direction	Determined by EA tilting angle	Determined by direction of M_{2y}
H_x Sensitivity	Insensitive	Change switching directions
H_y Sensitivity	Offset M_{1y} vs. torque loops	Making switching harder
Transition Route	$B \leftrightarrow D$ as $\theta_0 < 0$ and $A \leftrightarrow C$ as $\theta_0 > 0$	$A \leftrightarrow B$ as $M_{2x} < 0$; $C \leftrightarrow D$ as $M_{2x} > 0$
Advantages	Robust immunity to external fields with a fixed switching direction	Controllability of switching directions by fields and M_{2x}
Significance	(1) Demonstrate a new scheme to switch PMA films with high robustness (2) Develop a MRAM compatible structure to support two switching modes (3) Benefit for MRAM, multistate memory and spin logic applications	

Reviewers' comments:

Reviewer #3 (Remarks to the Author):

First of all, I would like to commend the authors' careful examination of their hypothesis and the detailed numerical simulations. With the possible tilting of easy axis, I agree that deterministic switching of perpendicular magnetization is now allowed by symmetry. I have several additional concerns:

(1) Based solely on simple symmetry argument, a tilting of perpendicular easy axis of a single layer ferromagnetic metal is enough to allow field-free switching. That is in the type-T configuration, shown in Fig. 1(d), but without the IMA layer. Even the current-induced in-plane Oersted field can potentially cause switching. Supplementary Information section 3 aims to investigate the role of interlayer coupling, and by turning the coupling off, the critical switching current seems higher. However, the numerical model does not seem to take into consideration the current-induced Oersted field or field-like torque.

(2) If easy axis tilting is important, can the authors estimate how much tilting of easy axis they got, e.g. from analyzing the offset in Fig. 2d? Without providing this quantitative information, it is hard to compare the experiments to the simulation.

(3) OoE data for Fig. 3c and Fig. 4c should be from the same data, but they look inconsistent.

Finally, my biggest concern is whether the paper is novel enough for Nature Communications. I agree the type-T configuration may be an alternative method to realize field-free magnetization switching. However, compared to the type-z configuration (also field-free with interlayer coupling) in their same sample, type-T requires an even larger current density. It further requires the tilting of easy axis, which is very tricky to get. The authors suggest that this may be from annealing in an external field (I suppose this has to be a tilted magnetic field.), but this was not carefully investigated. It is hard to judge how transferable this technique is if other groups want to repeat this experiment.

Responses to the 3rd Referee's Comments

First of all, we sincerely appreciate Review #3 for critical comments that have helped us to improve the clarity and quality of the revised manuscript. Now the symmetric analysis is in good agreement with experimental results. Below we provide point-to-point response to all the comments.

First of all, I would like to commend the authors' careful examination of their hypothesis and the detailed numerical simulations. With the possible tilting of easy axis, I agree that deterministic switching of perpendicular magnetization is now allowed by symmetry. I have several additional concerns:

Reply: We sincerely appreciate the positive comments and have checked correctness of the simulation results.

(1) Based solely on simple symmetry argument, a tilting of perpendicular easy axis of a single layer ferromagnetic metal is enough to allow field-free switching. That is in the type-T configuration, shown in Fig. 1(d), but without the IMA layer. Even the current-induced in-plane Oersted field can potentially cause switching. Supplementary Information section 3 aims to investigate the role of interlayer coupling, and by turning the coupling off, the critical switching current seems higher. However, the numerical model does not seem to take into consideration the current-induced Oersted field or field-like torque.

Reply: We sincerely appreciate the positive comments. It is true that the tilting of PMA EA can solely break the mirror-symmetry. Even so, a small tilting angle, for example 10° , is still not sufficient to deterministically switch a single PMA layer without external field. Torque can only pull the PMA layer in-plane in this case (Figure S11(a) and Figure R1(d)). However, this small tilting angle is sufficient to induce deterministic switching of a PMA layer in type *T* mode as shown in Figure S11(b) and also in Figure R2(c) and (d). In other words, *T*-type structure can reduce tilting requirement, compared with only switching a single tilted PMA layer.

Figure R1. (a) Dependence of M_{1z} on SOT as $A=0$. Large torque can only pull magnetization in-plane instead of switching. (b) Dependence of M_{2y} on SOT as $A=0$. Type- y switching is possible in this case. (c) Spin dynamics to switch the IMA layer with SOT (σ) being 0.6. (d) Spin dynamics to switch the IMA layer with σ being 0.8. Besides of switching of the IMA layer, it is clear that the PMA layer is pulled in-plane. Therefore, even though the PMA layer has a tilt easy axis (10° tilting angle), it cannot be switched by SOT in field-free condition.

Figure R2. Dependence of (a) M_{1z} and (b) M_{2y} on SOT with interlayer coupling of $A=0.6$ switched on. It is clear that both PMA with 10° tilt easy axis and IMA layer can be switched at the same critical SOT, indicating correlation of their spin dynamics. Subfigures (c) and (d) show spin dynamics of the PMA and IMA layers at $\sigma=0.6$ and $\sigma=0.8$. They both indicate the PMA layer is first pulled in-plane ($M_{1z} \sim 0$) and then switched toward -1 after the IMA layer switches from $M_{2y}=1$ to $M_{2y}=-1$. Comparing Figure R1 and R2, we can see Type- T structure can lower requirement on the tilting for field-free switching.

Basically, SOT and STT in Ta/CoFeB/MgO system have been extensively studied. It is widely recognized that the anti-damping-like torque plays dominant roles in the SOT-switching phenomenon instead of field-like torque or Oersted field, though the latter factors could help to reduce critical switching current [X. Zhang, et al. *Phys. Rev. B* **94** (2016) 174434; S. Fukami, et al. *Nat. Nanotechnol.* **11** (2016) 621-625]. Besides, “the effective field from field-like torque and Oersted field are parallel to the spin polarization (of the IMA layer) hence they will not influence the switching in a *qualitative* way” as stated by the 2nd referee. Anti-damping-like torque does matter in this case. Furthermore, the macrospin model based on anti-damping-like torque we adopted here has successfully reproduced all the features of Type-*T* and Type-*Z* modes, also indicating the minor or secondary influences of the Oersted field or field-like torques.

(2) If easy axis tilting is important, can the authors estimate how much tilting of easy axis they got, e.g. from analyzing the offset in Fig. 2d? Without providing this quantitative information, it is hard to compare the experiments to the simulation.

Reply: As suggested by the 3rd referee, by evaluating offset in *M-I* curves of Type-*T* mode, we can obtain the tilting angle in our case about 10° as shown in Figure R3 and Figure R4.

Figure R3. Simulated dependence of M_{1z} on SOT in Type- T mode. It is clear that the offset becomes gradually larger as the tilting angle of the PMA layer becomes larger. We can read out J_{c+} and J_{c-} . J_c is proportional to σ_c . Then we can obtain $J_c=(J_{c+}+J_{c-})/2$, $J_{ex}=(J_{c+}-J_{c-})/2$ and J_{ex}/J_c as shown in Figure R4. The ratio of J_{ex}/J_c monotonically increases with the tilting angle and can be used as marker to evidence the tilting angle of a real T -type system.

Figure R4. Simulated J_{ex}/J_c as a function of the tilting angle. They monotonically increase with each other. The dotted line indicated the experimental value retrieved from Figure 2(d), which means the tilting angle of the PMA layer in our T -type structure is around -10° .

(3) 0 Oe data for Fig. 3c and Fig. 4c should be from the same data, but they look inconsistent.

Reply: the data in Fig.3c and 4c come from two samples. In fact, to confirm these novel switching phenomena, we had systematically tested several tens of samples with different anisotropy energies and distinct shapes. All these samples suggested similar switching abilities and the field-free switching in type- T mode is revealed very robust. For example, Figure R5 show the switching data from a typical sample. Its switching behaviors both show the following characteristics. (1) Switching chirality is always fixed even as fields with opposite directions are applied. (2) No remarkable offset is observed when applying H_x . (3) In contrast, clear offset is observed when applying H_y . These data from the same sample follow the similar law with Figure 3c and 4c. After checking reproducibility by testing tens of samples with varied anisotropic strengths and shapes, we confirm almost every sample shows similar and robust switching. It indicates the Type- T switching is transferable. By showing different sets of data from different samples in the main text, we intend to show reproducibility among different samples.

Figure R5. The SOT switching loops of two different samples. They show almost the same switching characteristics of Type- T mode.

Finally, my biggest concern is whether the paper is novel enough for Nature Communications. I agree the type- T configuration may be an alternative method to realize field-free magnetization switching. However, compared to the type- z configuration (also field-free with interlayer coupling) in their same sample, type- T requires an even larger current density. It further requires the tilting of easy axis, which is very tricky to get. The authors suggest that this may be from annealing in an external field (I suppose this has to be a tilted magnetic field.), but this was not carefully investigated. It is hard to judge how transferable this technique is if other groups want to repeat this experiment.

Reply: We argue in this manuscript that not only Type- Z but also Type- T mode could be both achieved in this CoFeB/Ta/CoFeB T -type coupled system with/without external fields, which could greatly expand functions and controllability of a SOT device with this T -type magnetic structure. It is the impact of this work in applications. Physically, Type- T mode provides an alternative possibility to switch PMA layer by switching another coupled IMA layer, which uncovers a new switching dynamics for PMA materials. The novelty both in applications and physics endows impact to this paper, we think.

Regarding critical switching current density, switching the PMA layer in Type- T mode is realized by switching the coupling IMA layer. However, different from the Type- y mode [S. Fukami, et al. *Nat. Nanotechnol.* 11 (2016) 621-625], switching the IMA layer in Type- T mode could be easier than switching a single IMA layer in Type- y mode because coupling with a PMA layer can give the IMA layer an initial kick-off angle which is indispensable and usually provided by thermo-fluctuation in classic type- y mode. Thus Type- T mode can at least theoretically provide a more efficient

way to switch both PMA and IMA layers than Type-Z mode (as shown in Figure 1 (e)-(h) of the main text) and Type-y mode (as shown in Figure R1(b) and R2(b)).

However, we think in our main text, the critical current density of Type- T mode should not be directly used to be compared with that of Type-Z mode, because Type- T mode realized a 100 % switching while Type-Z mode only realized less than 70% switching in field-free condition. Furthermore, only in the presence of external bias field could Type Z mode also realize 100% switching as Type- T mode. The critical switching current in Type- T mode in this case is higher than that of Type-Z mode by a factor of 2. However, Type- T switching is much sharper. Due to absence of applied field for full switching, we think Type- T mode can be at least as efficient as Type-Z mode, if not more efficient.

Regarding transferability, we have shown that an ordinary high-temperature annealing process in bias field as introduced in the manuscript could lead to the desired magnetic anisotropy. Then we further realized Type- T (as shown above in Figure R5) and Type- z modes in many samples with the T -type anisotropy. This reproducibility itself indicates transferability of this technique.

Though there are still optimization margins for both Type- T and Type-Z modes, this unique magnetic structure and versatile SOT-switching modes could give birth to more complex and multi-functional spintronic devices such as spin logics and multi-state memories, which is the main contribution of this work.

Finally, the above comments and suggestions from the referees are highly appreciated again.